# The role of B cells in immune cell activation in polycystic ovary syndrome

Angelo Ascani[1†], Sara Torstensson[2†], Sanjiv Risal[2], Haojiang Lu[2], Gustaw Eriksson[2], Congru Li[2], Sabrina Teschl[1], Joana Menezes[2], Katalin Sandor[2], Claes Ohlsson[3,4], Camilla I Svensson[2], Mikael CI Karlsson[5], Martin Helmut Stradner[1], Barbara Obermayer-Pietsch[1*‡], Elisabet Stener-Victorin[2*‡]

[1]Department of Internal Medicine, Medical University of Graz, Graz, Austria; [2]Department of Physiology and Pharmacology, Karolinska Institute, Stockholm, Sweden; [3]Centre for Bone and Arthritis Research, Department of Internal Medicine and Clinical Nutrition, Sahlgrenska Academy, University of Gothenburg, Gothenburg, Sweden; [4]Department of Drug Treatment, Sahlgrenska University Hospital, Region Västra Götaland, Gothenburg, Sweden; [5]Department of Microbiology, Tumor and Cell Biology, Karolinska Institute, Stockholm, Sweden

*For correspondence:
barbara.obermayer@
medunigraz.at (BO-P);
elisabet.stener-victorin@ki.se
(ES-V)

†These authors contributed
equally to this work
‡These authors also contributed
equally to this work

Reviewing Editor: Larisa V
Suturina, Scientific Center
for Family Health and Human
Reproduction problems, Russian
Federation

## Abstract

Variations in B cell numbers are associated with polycystic ovary syndrome (PCOS) through unknown mechanisms. Here, we demonstrate that B cells are not central mediators of PCOS pathology and that their frequencies are altered as a direct effect of androgen receptor activation. Hyperandrogenic women with PCOS have increased frequencies of age-associated double-negative B memory cells and increased levels of circulating immunoglobulin M (IgM). However, the transfer of serum IgG from women into wild-type female mice induces only an increase in body weight. Furthermore, RAG1 knockout mice, which lack mature T- and B cells, fail to develop any PCOS-like phenotype. In wild-type mice, co-treatment with flutamide, an androgen receptor antagonist, prevents not only the development of a PCOS-like phenotype but also alterations of B cell frequencies induced by dihydrotestosterone (DHT). Finally, B cell-deficient mice, when exposed to DHT, are not protected from developing a PCOS-like phenotype. These results urge further studies on B cell functions and their effects on autoimmune comorbidities highly prevalent among women with PCOS.

## Editor's evaluation

This manuscript provides the fundamental findings regarding the alteration of B cell frequencies and functionality, caused by the activation of androgen receptor. Based on the compelling strength of evidence, the manuscript presents significant results improving the current understanding of PCOS and associated disorders. The manuscript will be of interest to medical biologists, gynaecologists, and endocrinologists.

## Introduction

Polycystic ovary syndrome (PCOS) is the single most common endocrine-metabolic disorder affecting 5–18% of women in reproductive age worldwide (*Joham et al., 2022*). As a multifactorial disorder with no clearly defined etiology, PCOS is diagnosed based on the criteria of hyperandrogenism, oligo-anovulation, and polycystic ovarian morphology (PCOM) (*Joham et al., 2022*). The syndrome is characterized by chronic low-grade inflammation and potential autoimmune sequelae (*Gaberšček et al., 2015*), further aggravated by obesity. Indeed, women affected with PCOS are at increased risk for type 2 diabetes mellitus (T2D) (*Azziz, 2018*) with a number of studies indicating also a higher

**eLife digest** Polycystic ovary syndrome is a lifelong condition associated with disrupted hormone levels, which affects around 15-20% of women. Characterised by increased levels of male sex hormones released by ovaries and adrenal glands, the condition affects menstrual cycles and can cause infertility and diabetes.

Alongside the increase in male sex hormones, changes in the number of B cells have recently been observed in polycystic ovary syndrome. B cells produce antibodies that are important for fighting infection. However, it is thought that they might aggravate the condition by releasing antibodies and other inflammatory molecules which instead attack the body. It remained unclear whether changes in the B cell numbers were a result of excessive hormone levels or whether the B cells themselves were responsible for increasing the levels of male sex hormones.

Ascani et al. showed that exposing female mice to excess male sex hormones leads to symptoms of polycystic ovary syndrome and causes the same changes to B cell frequencies as observed in women. This effect was prevented by simultaneously treating mice with a drug that blocks the action of male sex hormones. On the other hand, transferring antibodies from women with polycystic ovary syndrome to mice led to greater body weight and variation in B cell numbers. However, it did not result in clear symptoms of polycystic ovary syndrome. Furthermore, mice without B cells still developed symptoms when exposed to male sex hormones, showing that B cells alone are not solely responsible for the development of the condition.

Taken together, the experiments show that B cells are not central mediators of polycystic ovary syndrome and the variation in their numbers is due to excess male sex hormones. This raises the question of whether B cells are an appropriate target for the treatment of this complex condition and paves the way for studies on how other immune cells are altered by hormones. Future work should also investigate how B cell function affects symptoms associated with polycystic ovary syndrome, given the association between antibody transfer and weight gain in mice.

prevalence of autoimmune thyroid disease (AITD), and particularly hypothyroidism (*Gaberšček et al., 2015*). In support of this observation, serological parameters of autoimmunity, such as anti-histone and anti-double-stranded DNA antibodies, are relatively high among these women (*Hefler-Frischmuth et al., 2010*). However, attempts to identify an autoimmune cause of PCOS have been uninformative (*Gleicher, 1998*; *Mobeen et al., 2016*; *Petríková et al., 2010*). Hyperandrogenemia is a hallmark feature of PCOS that plays a key role in the pathophysiology and seems to be directly related to disease severity (*Joham et al., 2022*; *Diamanti-Kandarakis and Dunaif, 2012*). Although chronic inflammation and altered immune function have been proposed to play a role in the pathogenesis of PCOS and T2D, it remains unknown whether the observed immune responses and autoimmune alterations in women with PCOS are a cause or consequence of hyperandrogenemia (*Hu et al., 2020*).

The notion of B cells exacerbating metabolic disease has been known for over a decade, both in the pathogenesis of diabetes (*Deng et al., 2016*) as well as in obesity-associated insulin resistance (*Winer et al., 2011*). Although being an extreme model, biological variations in mu heavy chain knockout mice (MuMt⁻; B$^{null}$), which fail to produce mature B cells, have contributed to this knowledge (*Winer et al., 2011*). Notably, muMt⁻ mice reconstituted with B cells derived from mice with diet-induced obesity (DIO) develop an impaired glucose tolerance. However, when transferring immunoglobulin G (IgG) from DIO mice to muMt mice, systemic inflammatory changes were noticeable only when the recipient mice were exposed to an high-fat diet, suggesting that metabolic effects stemming from B cells may require exposure to a prior stimulus as a determinant of reaction, via induced conditioning or induction of target autoantigens (*Winer et al., 2011*).

These observations led us to investigate whether the hyperandrogenic hormonal milieu in PCOS could have a predominant role in B cell fate as the androgen receptor (AR) is expressed both in immune organs as well as on precursors and some mature immune cells, potentially implicating various levels of susceptibility (*Gubbels Bupp and Jorgensen, 2018*). Testosterone is also an indirect regulator of the cytokine B cell activating factor (BAFF), also known as TNFSF13B, an essential survival factor for B cells (*Wilhelmson et al., 2018*), which has been shown to be increased in women affected with PCOS (*Xiao et al., 2019*). This may be a plausible candidate mechanism as excessive BAFF levels

allow for the survival of autoreactive cells and autoantibody production (*Vincent et al., 2014*). Recent research has shown that the proportions and activity of peripheral B cells in women with PCOS are increased (*Xiao et al., 2019*), though it remains unclear whether B cells alone are the main inflammatory drivers of PCOS pathogenesis and whether hyperandrogenemia through AR activation may lead to the acquisition of their unique characteristics. When examining potential autoreactive B cells in PCOS, it is important and clinically meaningful to discern the role of androgen exposure alone on the immune system from that of obesity-deriving inflammation. Plasma of individuals with obesity has been shown to be enriched in IgG antibodies with anti-self-reactivity, which have been positively associated with blood frequencies of double negative (DN) B cells, considered the most pro-inflammatory B cell subset (*Frasca et al., 2021*). How androgen exposure alone, through androgen receptor activation, affects B cells and their function in a PCOS-like mice model exhibiting a reproductive and metabolic phenotype (*Risal et al., 2019*) remains unknown.

In this study, we investigate the role of B cells in the underlying inflammation of PCOS by assessing the effect of hyperandrogenism on B cell populations and whether B cells are contributing to the pathology. For this purpose, in line with previous data coupling increased B cells numbers (CD19[+]) with PCOS, we first aimed to define which main B cell lineages are affected in hyperandrogenic women with PCOS. Next, we assessed whether B cells with self-reactive potential may have a causal effect on both the development of a PCOS-like phenotype, including metabolic dysfunction, and the immune profile in mice by transferring IgG from women with PCOS. To further study whether B cell frequencies are altered in reproductive, metabolic, and immunological tissues, major variations of B cell subsets were analyzed in a dihydrotestosterone (DHT)-induced PCOS-like mouse model. Furthermore, we investigated whether these DHT-induced alterations are a result of androgen receptor activation by simultaneous administration of flutamide, an androgen receptor antagonist. Finally, we questioned whether B cell-deficient muMt[-] mice are protected from developing PCOS traits when exposed to DHT.

Here, we demonstrate that AR activation is a direct modulator of B cell frequencies in PCOS pathogenesis. We show that the transfer of circulating IgG from women with PCOS disrupts B cell proportions and causes an increased body weight and sex steroid imbalance in female WT mice. Collectively, our data support a model wherein activation of B cells promotes the development of a PCOS-like metabolic phenotype in mice. We also suggest caution toward therapeutically targeting CD19[+] cells as DHT-exposed B cell-deficient mice develop a PCOS-like phenotype, showing that a lack of B cells is not protective and reiterates the need for broader studies on alterations of the immune system within the complex hormonal frame of PCOS, including activation of T cells and tissue-resident immune cells.

## Results

### Androgens are associated with altered B cell frequencies and immunoglobulin M increase in women with PCOS

As alterations of B cell frequencies have previously been shown in women with PCOS (*Xiao et al., 2019*), we first characterized main B cell lineages and subpopulations based on pan B cell surface marker CD19 in the serum of 15 hyperandrogenic women with PCOS and of 22 women without PCOS (controls). Women with PCOS fulfilled all three Rotterdam Criteria, displaying oligo-/amenorrhea, hirsutism, and PCOM. Women with PCOS were younger than controls, with median ages of 26 and 36, respectively, with no difference in body mass index (BMI), but with significantly higher total testosterone and androstenedione, elevated total triglycerides, and reduced HDL-cholesterol (*Table 1*). CD19[+] B cell memory populations were phenotypically analyzed based on surface markers IgD and CD27. Our initial assessment showed a remodeling of B cell repertoire in women with PCOS compared to controls. The frequency of age-associated DN B memory cells lacking surface expression of CD27 and IgD was significantly higher in women with PCOS (*Figure 1a*), with declined 'innate-like' unswitched CD27[+]IgD[+] B memory cells (*Figure 1b*). While naïve B cells populations did not differ among study groups (*Figure 1c*), switched CD27[+] IgD[−] were increased in women with PCOS (*Figure 1d*), which may directly affect unswitched B cells frequency variance. We did not find direct evidence of activation of DN B cells among women affected by PCOS. Analysis of surface marker CD38, generally expressed on antibody-secreting plasma cells, proved similar values in both groups (*Figure 1e*). Expression of CD86, a co-stimulatory molecule that usually is upregulated following

**Table 1.** Clinical characteristics of women with polycystic ovary syndrome (PCOS) and women without the syndrome used for characterization of main B cell lineages and subpopulations based on pan B cell surface marker CD19.

| | Controls (n = 22) | PCOS (n = 15) | p-Value |
|---|---|---|---|
| Age (years) | 36.3 (21-50) | 26.4 (24-38) | **0.003** |
| | | | |
| *Anthropometry* | | | |
| BMI (kg/m$^2$) | 22.18 (18.31–32.15) | 24.39 (19.16–39.89) | 0.128 |
| Waist-to-hip-ratio | 0.8 (0.74–0.91) | 0.82 (0.71–0.92) | 0.237 |
| | | | |
| *Endocrine measure* | | | |
| Free testosterone (ng/mL) | 0.25 (0,06–0.49) | 0.43 (0.02–0.88) | **0.037** |
| Total testosterone (ng/mL) | 0.86 (0.8–1.74) | 1.49 (0.07–3.05) | **0.037** |
| Free androgen index (FAI) | 1.2 (0.22–2.48) | 2.6 (0.11–20.34) | **0.003** |
| Androstenedione (ng/mL) | 2.8 (1.17–5.99) | 3.9 (2.27–6.53) | **0.003** |
| | | | |
| *Metabolic measures* | | | |
| Cholesterol (mg/dL) | 177 (128–217) | 180 (135–214) | 0.469 |
| HDL (mg/dL) | 67 (46–87) | 51 (36–90) | **0.001** |
| LDL (mg/dL) | 97.3 (45.8–115.8) | 108.6 (78.6–160.4) | **0.049** |
| Triglycerides (mg/dL) | 65.5 (44-97) | 82 (56–124) | **0.011** |
| Glucose (mg/dL) | 88 (76–104) | 91 (77–111) | 0.491 |

Data are median ± range. Comparisons between groups were made using Mann–Whitney *U*-test.
BMI, body mass index; HDL, high-density lipoprotein; LDL, low-density lipoprotein.

activation of B cells and in turn can activate T cells, did not differ significantly either (**Figure 1f**). When assessing circulating serum antibodies, immunoglobulin M (IgM) were higher in hyperandrogenic women with PCOS (**Figure 1g**) exhibiting high testosterone and increased free androgen index (**Figure 1h and i**) compared to controls with similar BMI (**Figure 1j**). Interestingly, no differences were noted for circulating IgG while IgA titers were lower in women with PCOS (**Figure 1—figure supplement 1**).

These data support the hypothesis that women with PCOS and hyperandrogenemia have an altered B cell frequency linked to alterations in IgM antibody production. However, higher disease activity was not explained by increased double-negative (DN) B lymphopoiesis.

## Transfer of human-derived IgG antibodies results in increased body weight in WT female mice

Clusters of pro-inflammatory age-associated DN B memory cells lacking surface expression of CD27 and immunoglobulin D (IgD) have been associated with plasma cell differentiation fate, and while not increasing significantly in numbers, produce higher amounts of IgG on a per cell basis relative to switched memory B cells (**Jenks et al., 2018**). Hence, to assess whether PCOS may have an underlying autoimmunological effector component, we investigated a possible role for IgG in PCOS systemic inflammation. IgG antibody extracted from serum of four women with PCOS diagnosed as phenotype A fulfilling all three Rotterdam Criteria, displaying oligo-/amenorrhea, hirsutism, and PCOM and of healthy controls (**Table 2**), were purified and pooled, then transferred intraperitoneally (i.p.) into wild-type (WT) mice. Following the same procedure, IgG deriving from four hormonally healthy women was equally purified, then pooled and transferred into similar age and weight-matched WT mice. Both

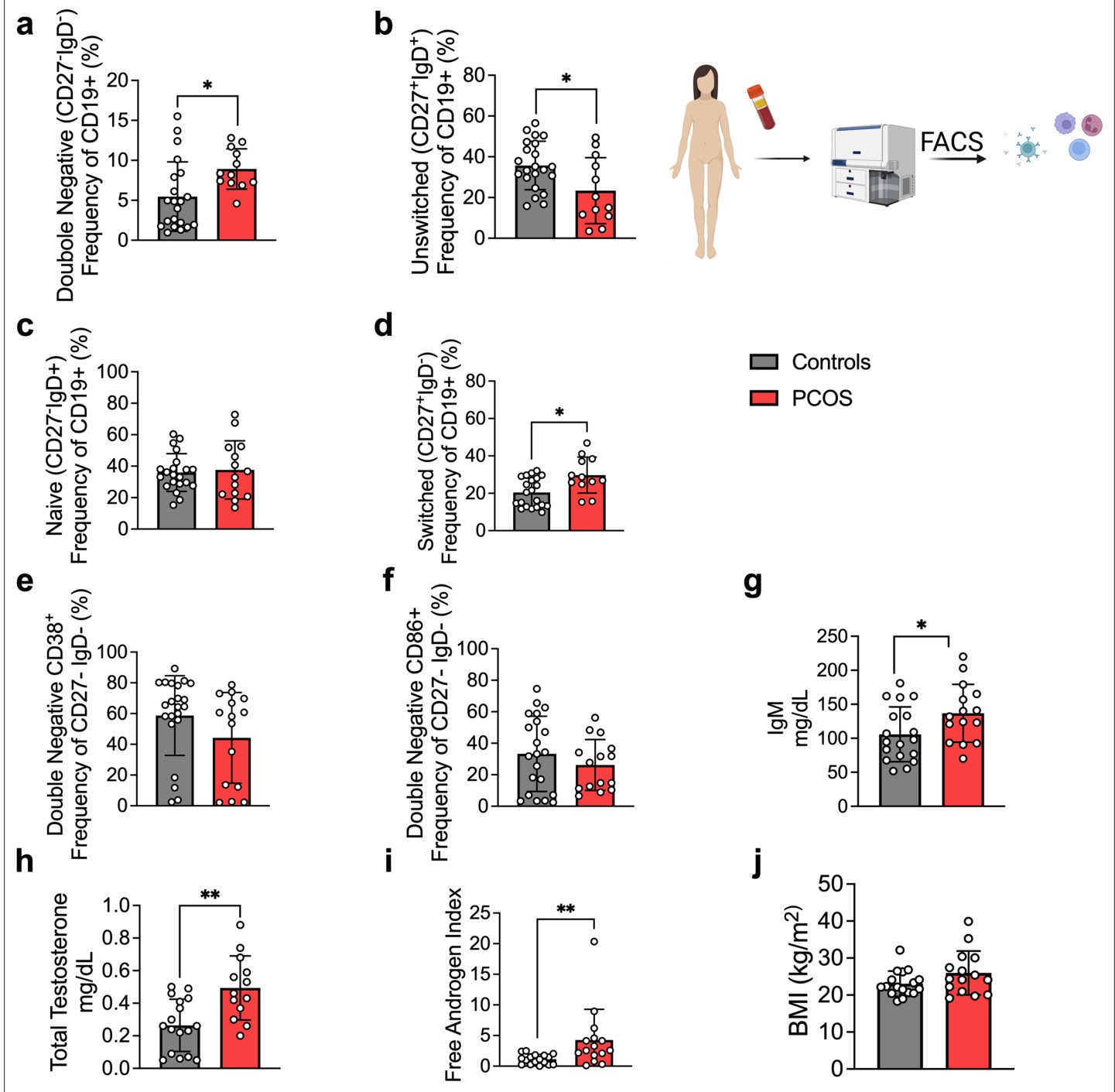

**Figure 1.** B cell frequencies and immunoglobulin M (IgM) variations in women with polycystic ovary syndrome (PCOS). (**a**) Total CD19+ double-negative (DN) B cells (CD27- IgD-). (**b**) Total unswitched B cells (CD27+IgD+ ). (**c**) Total naive B cells (CD27- IgD+). (**d**) Total switched B cells (CD27+IgD-). (**a–d**) Total CD19+ populations (controls n = 22; PCOS n = 15). (**e, f**) Expression on double-negativeDN B cells respectively of the surface markers CD38 and CD86. (**g**) Circulating IgM titers. (**h**) Total testosterone. (**i**) Free androgen index (FAI). (**j**) Body mass index (BMI). (**g–j**) Controls n = 18; PCOS n = 15. All bars indicate means, error bars SD, circles represent human individuals. In the case of missing values due to lack of measurement, individuals were excluded from the analysis report for that variable. Unpaired Student's t-test for analysis of naive, unswitched, and DN CD86+ B cells, total testosterone, and BMI. Mann–Whitney test for all other B cell frequencies, antibody titers, and FAI. *p<0.05, **p<0.01, ***p<0.001.

The online version of this article includes the following figure supplement(s) for figure 1:

**Figure supplement 1.** Circulating antibody titers in women with and without polycystic ovary syndrome (PCOS).

**Table 2.** Clinical characteristics of immunoglobulin G (IgG) donors, women with polycystic ovary syndrome (PCOS) and women without the syndrome.

| | Controls (n = 4) | PCOS (n = 4) | p-Value |
|---|---|---|---|
| Age (years) | 27 (22–31) | 25 (23–35) | >0.999 |
| | | | |
| *Anthropometry* | | | |
| BMI (kg/m$^2$) | 26 (19.4–29.8) | 25 (21.3–28.2) | >0.999 |
| Waist-to-hip-ratio | 1 (0.79–0.91) | 1 (0.70–0.91) | 0.857 |
| | | | |
| *Endocrine measure* | | | |
| LH (mU/mL) | 7 (4.40–11.40) | 10 (5.19–38.60) | 0.343 |
| FSH (mU/mL) | 4 (2.74–6.91) | 7 (5.59–8.61) | **0.057** |
| Progesterone (ng/mL) | 10 (0.20–13.60) | 1 (0.60–1.05) | 0.343 |
| Free testosterone (ng/mL) | 2 (0.84–2.68) | 3 (0.29–3.03) | 0.685 |
| Total testosterone (ng/mL) | 0 (0.17–0.40) | 0 (0.30–0.40) | 0.228 |
| Androstenedione (ng/mL) | 3.11 (1.21–4.56) | 4 (1.98–4.69) | 0.685 |
| SHBG (nmol/L) | 63 (52.8–88.8) | 62 (43.9–105) | 0.952 |
| Free androgen index (FAI) | 0 (0.3–0.6) | 1 (0.5–0.7) | 0.171 |
| AMH (ng/mL) | 4 (2.40–4.66) | 8 (4.97–9.96) | **0.028** |
| | | | |
| *Metabolic measures* | | | |
| Cholesterol (mg/dL) | 160 (132–184) | 153 (146–172) | 0.686 |
| HDL (mg/dL) | 77 (42–80) | 62 (49–71) | 0.343 |
| LDL (mg/dL) | 75 (66–89) | 74 (67–107) | 0.828 |
| Triglycerides (mg/dL) | 71 (43–91) | 74 (68–82) | 0.885 |

Data are median ± range. Comparisons between groups were made using Mann–Whitney *U*-test.
AMH, anti-Müllerian hormone; HDL, high-density lipoprotein; LDL, low-density lipoprotein; SHBG, sex hormone-binding globulin; FSH, follicle-stimulating hormone; LH, luteinizing hormone.

groups of donors were age-homogeneous with no significant differences in BMI or circulating androgen levels (*Table 2*). Among the recipient mice, there were no differences in ovulatory cycles between controls (*Figure 2a*) and mice receiving PCOS IgG (*Figure 2b*). Three weeks post IgG transfer, mice receiving IgG from women with PCOS increased in body weight compared to controls (*Figure 2c*). Body composition assessment showed no difference in proportion of fat or lean mass between the groups (*Figure 2d*). Interestingly, as an effect of human PCOS IgG transfer, recipient mice had also altered subsets of B lymphocytes in blood, ovary, and visceral adipose tissue (VAT). Circulating DN B memory cells were increased (*Figure 2e*) while blood-naïve cells were reduced (*Figure 2f*), resembling the B cell distribution described in donor women with PCOS. Among the DN B cells, DN1 CD21$^+$ subset was the main circulating subpopulation in the blood of mice that received IgG from women with PCOS (*Figure 2g*). VAT tissue had higher frequencies of effector IgM$^+$IgD$^+$CD27$^+$ 'double positive' unswitched B cells (*Figure 2h*) while activated switched IgM$^+$IgD-CD27$^+$ were increased in ovarian tissue (*Figure 2i*). Analyzing circulating sex steroids in these mice, estrogens were altered with an increase in estrone (*Figure 2j*) and a trend of higher estradiol (*Figure 2k*) with no difference in androgens or progesterone (*Figure 2l–n*).

As B cell functions are influenced by other lymphocyte populations, especially T cells, and vice versa, we aimed to assess whether T cells are modulating this inflammatory effect deriving from IgG-induced

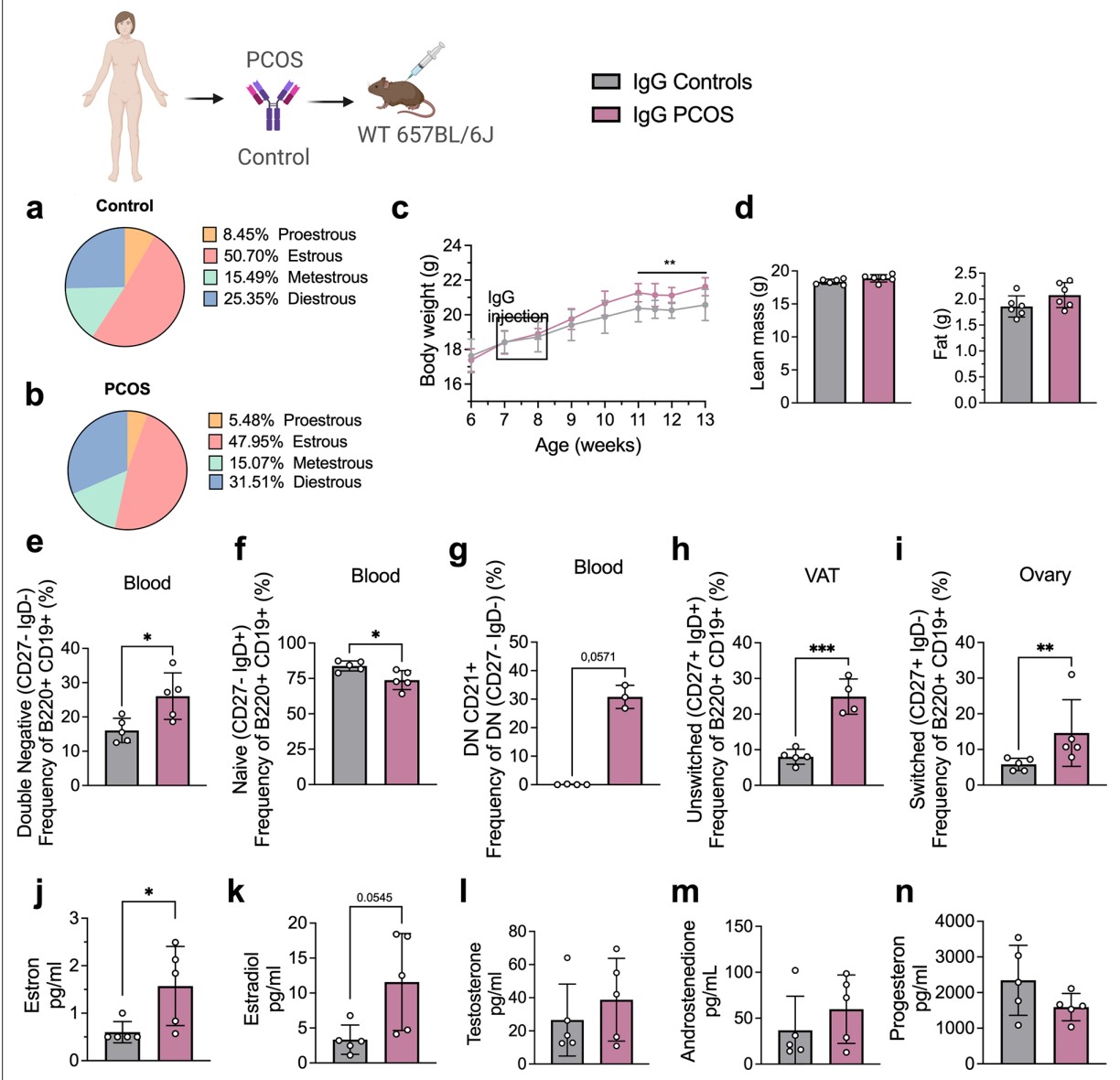

**Figure 2.** Immunoglobulin G (IgG) transfer to wild-type (WT) mice. (**a**) Estrous cycles in mice receiving control IgG. (**b**) Estrous cycles in mice receiving polycystic ovary syndrome (PCOS) IgG. (**c**) Weekly body weight (BW) recordings. (**d**) EcoMRI results for body fat and lean mass composition. (**e**) Double-negative (DN) B cells in blood. (**f**) Naive B cells in blood. (**g**) DN CD21+ B cells in blood. (**h**) Unswitched B cells in visceral adipose tissue (VAT). (**i**) Switched B cells in ovary. (**j**) Testosterone. (**k**) Androstenedione. (**l**) Estradiol. (**m**) Estron. (**n**) Progesterone. (**a–n**) IgG donors; controls n = 4, PCOS n = 4 (see *Table 2*). Mice receiving control IgG (n = 6), mice receiving PCOS IgG (n = 6). All bars indicate means, circles represent individual mice. In the case of missing values due to lack of measurement, mice were excluded from the analysis report for that variable. Unpaired Student's *t*-test for analysis of EchoMRI results and all B cell frequencies except DN CD21+ and switched; two-way ANOVA with Sidak's post hoc test for analysis of weekly BW recordings; Mann–Whitney test for analysis of DN CD21+, switched, estron, testosterone, and androstenedione, Welch's *t*-test for analysis of estradiol and progesterone. *p<0.05, **p<0.01, ***p<0.001.

disease. Human IgG deriving from PCOS and control cohorts was first purified, then pooled into separate groups, and transferred into 10-week-old Rag1 KO$^{-/-}$ mice, which lack mature T- and B cells. Three weeks post i.p. IgG transfer, RAG1 KO$^{-/-}$ mice failed to develop any PCOS-like phenotype, contrasting previous transfer into WT mice, suggesting that the pathophysiological mechanism inducing immune and metabolic disruption such as body weight alteration may necessarily involve other lymphocytes to fully promote impairment of metabolic parameters.

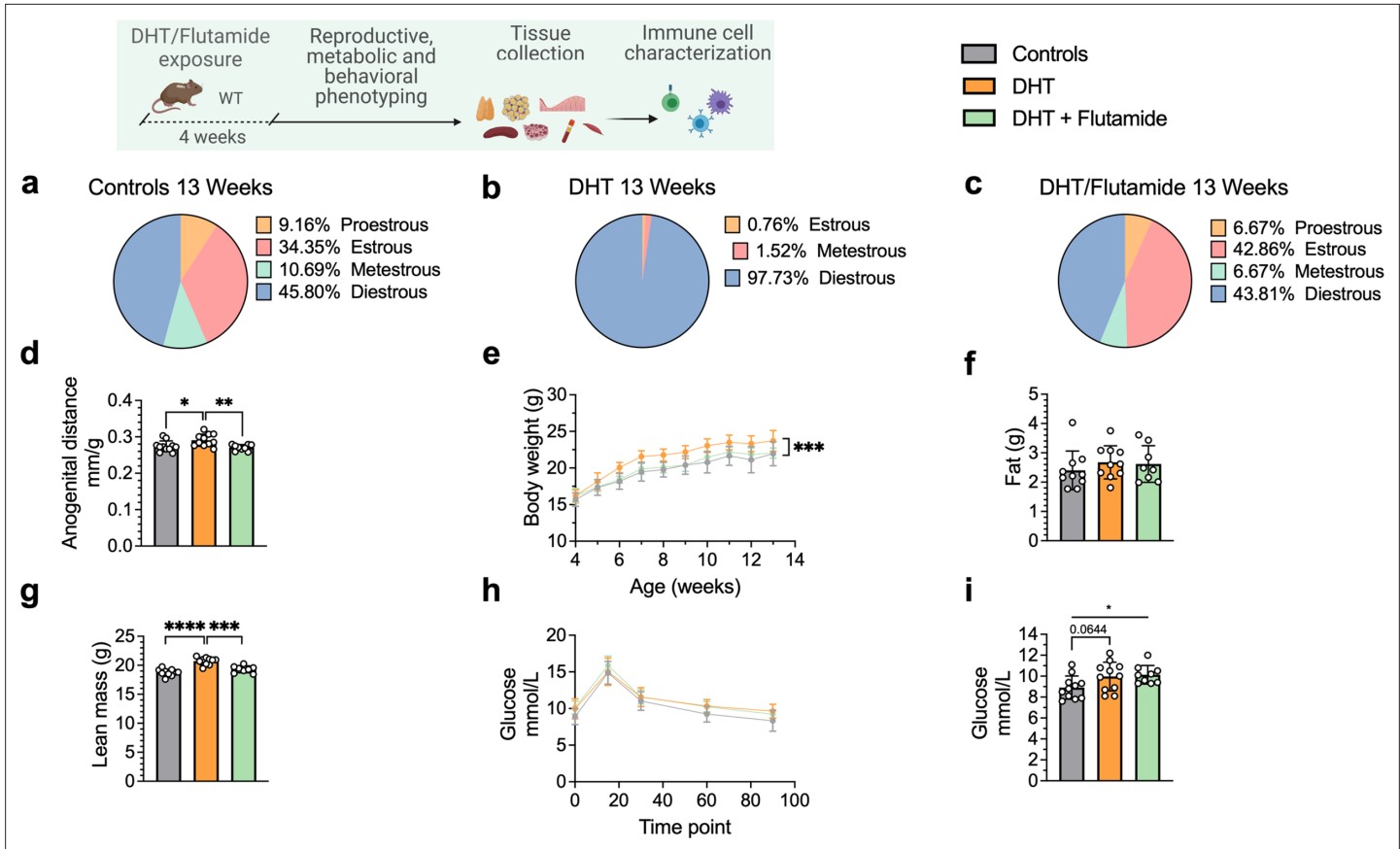

**Figure 3.** Dihydrotestosterone (DHT)-induced polycystic ovary syndrome (PCOS)-like mouse model phenotypic study at 13 weeks of age. (**a**) Estrous cycles in WT control mice. (**b**) Estrous cycles in mice receiving DHT pellet implant. (**c**) Estrous cycles in mice receiving DHT pellet and flutamide implant. (**d**) Normalized anogenital distance. (**e**) Weekly body weight (BW). (**f**) EchoMRI record of fat body composition. (**g**) EchoMRI record of lean body composition. (**h**) Oral glucose tolerance test (OgTT). (**i**) Fasting glucose. (**a–i**) WT control mice (n = 11), mice receiving DHT pellet implant (n = 11), mice receiving DHT pellet, and flutamide implant (n = 10). All bars indicate means, circles represent individual mice. In the case of missing values due to lack of measurement, mice were excluded from the analysis report for that variable. Unpaired Student's *t*-test for analysis of anogenital distance difference between groups, as well as EchoMRI results and fasting glucose; two-way ANOVA with Sidak's post hoc test for analysis of weekly BW recordings and blood glucose throughout the study. *p<0.05, **p<0.01, ***p<0.001.

## Altered B cell frequencies are replicated in a DHT-induced PCOS-like mouse model and seen in reproductive, metabolic, and immunological tissues

To investigate androgen-mediated regulation of B cell phenotypes, in particular DN B memory cells as well as circulating antibody titers, in tissues other than blood, we used the well-established peri-pubertal DHT-induced PCOS mouse model (*Stener-Victorin et al., 2020*). Peripubertal female mice were subcutaneously implanted with a silastic pellet containing 4 mm of DHT and develop PCOS-like traits with reproductive and metabolic dysfunction without increase in fat mass (*Xue et al., 2018*). Control mice received an empty, blank implant. To investigate whether any phenotypic differences are driven by AR activation, a third group received, in addition to the DHT implant, a continuously releasing flutamide pellet, an AR antagonist. Two separate cohorts of these three experimental groups were phenotypically characterized at 13 and 16 weeks of age, respectively.

After 4 weeks of continuous DHT exposure, mice had developed a reproductive PCOS-like phenotype, exhibiting disrupted estrous cycles, arrested in diestrus or metestrus (*Figure 3a and b*), and longer anogenital distance (*Figure 3d*). Co-treatment with flutamide prevented the development of these phenotypes (*Figure 3c and d*). DHT-exposed mice gained body weight (*Figure 3e*), not in fat mass (*Figure 3f*) but rather by increased lean mass (*Figure 3g*) compared to controls and mice co-treated with flutamide. At 12 weeks of age, no clear sign of impaired glucose homeostasis was

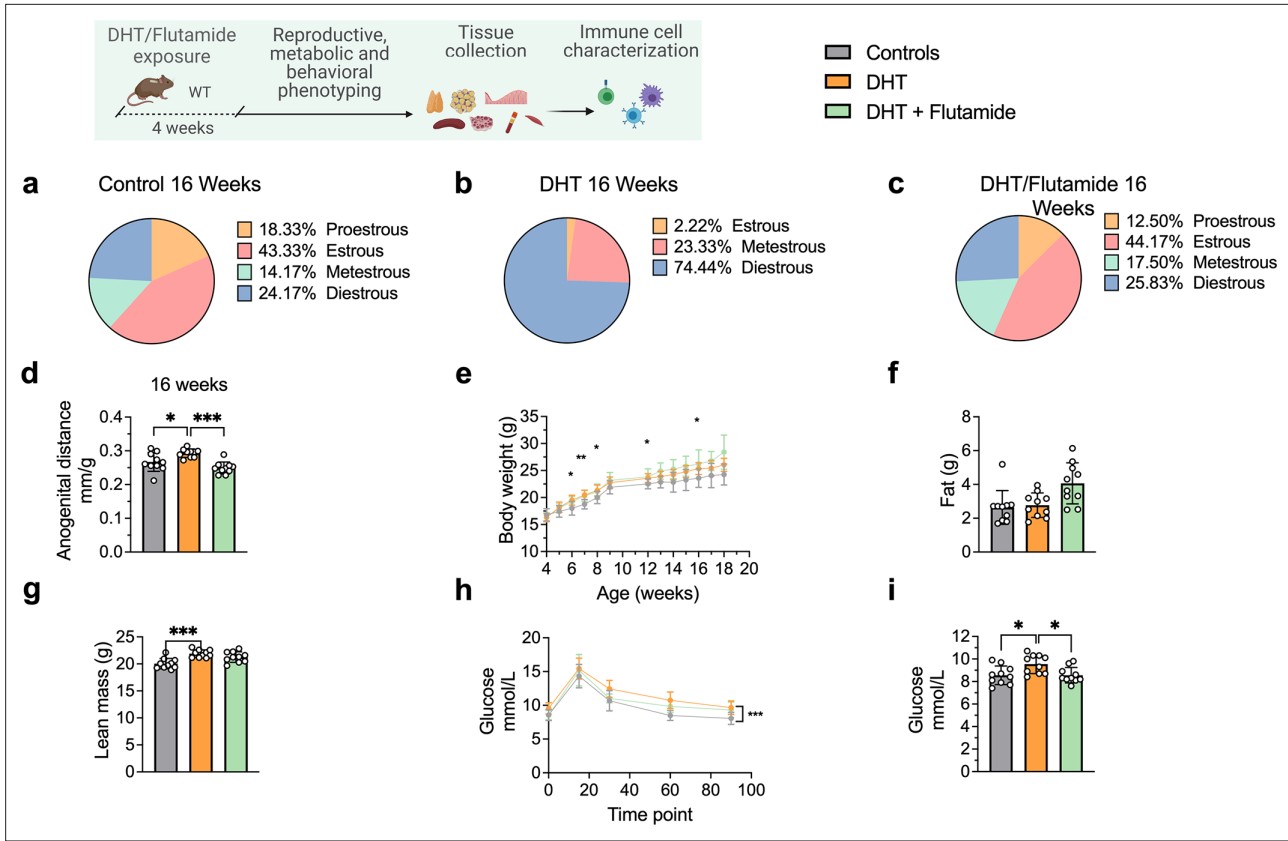

**Figure 4.** Dihydrotestosterone (DHT)-induced polycystic ovary syndrome (PCOS)-like mouse model phenotypic study at 16 weeks of age. (**a**) Estrous cycles in WT control mice. (**b**) Estrous cycles in mice receiving DHT pellet implant. (**c**) Estrous cycles in mice receiving DHT pellet and flutamide implant. (**d**) Anogenital distance normalized to body weight (BW). (**e**) Weekly BW. (**f**) EchoMRI record of fat body composition. (**g**) EchoMRI record of lean body composition. (**h**) Oral glucose tolerance test (OgTT). (**i**) Fasting glucose. (**a–i**) WT control mice (n = 10), mice receiving DHT pellet implant (n = 10), mice receiving DHT pellet and flutamide implant (n = 10). All bars indicate means, circles represent individual mice. In the case of missing values due to lack of measurement, mice were excluded from the analysis report for that variable. Unpaired Student's *t*-test for analysis of anogenital distance difference between groups, as well as EchoMRI results and fasting glucose; two-way ANOVA with Sidak's post hoc test for analysis of weekly BW recordings and blood glucose throughout the study. *p<0.05, **p<0.01, ***p<0.001.

noted during oral glucose tolerance test (oGTT) compared to controls (*Figure 3h*), although DHT-exposed mice display a trend of higher fasting glucose (*Figure 3i*).

At 16 weeks of age, DHT-exposed mice with a PCOS-like phenotype have an equally disrupted estrous cycle compared to controls (*Figure 4a*), arrested in diestrus or metestrus (*Figure 4b*) and longer anogenital distance (*Figure 4d*). These effects were prevented by co-treatment with flutamide as in previous experiment (*Figure 4c and d*). DHT-exposed mice weighed more (*Figure 4e*), an effect due to higher lean mass (fat mass, *Figure 4f*; lean mass, *Figure 4g*), and had impaired glucose homeostasis, displaying an overall impaired glucose uptake during oGTT (*Figure 4h*) and a higher fasting glucose and compared to controls (*Figure 4i*).

Frequencies of B memory cells in 13-week-old DHT-exposed mice were disrupted compared to controls, particularly in blood. Circulating CD19+ DN memory cells were lower compared to controls (*Figure 5a*). CD19+-naïve B cells were increased in the blood of DHT-exposed mice (*Figure 5b*). When analyzing B cell distribution at 20 weeks of age, CD19+ DN B cells were increased in the spleen of DHT-exposed mice (*Figure 5c*) while frequencies of naïve B cells were decreased (*Figure 5d*), an effect that was reversed when co-treated with flutamide. Overall, ovarian tissue was the most affected tissue. DHT-exposed mice had decreased proportions of both DN and naïve B cells within the ovaries (*Figure 5e and f*), a similar trend as seen in blood-deriving cells of mice at 13 weeks of age. Ovaries of DHT-exposed mice were characterized by an increased frequency of IgM+IgD+CD27+ 'double positive' unswitched B cells (*Figure 5g*). Among the DN cells, DHT-exposed mice displayed a trend, suggesting an increase of DN CD21+ populations in the ovaries compared with controls and mice co-treated with

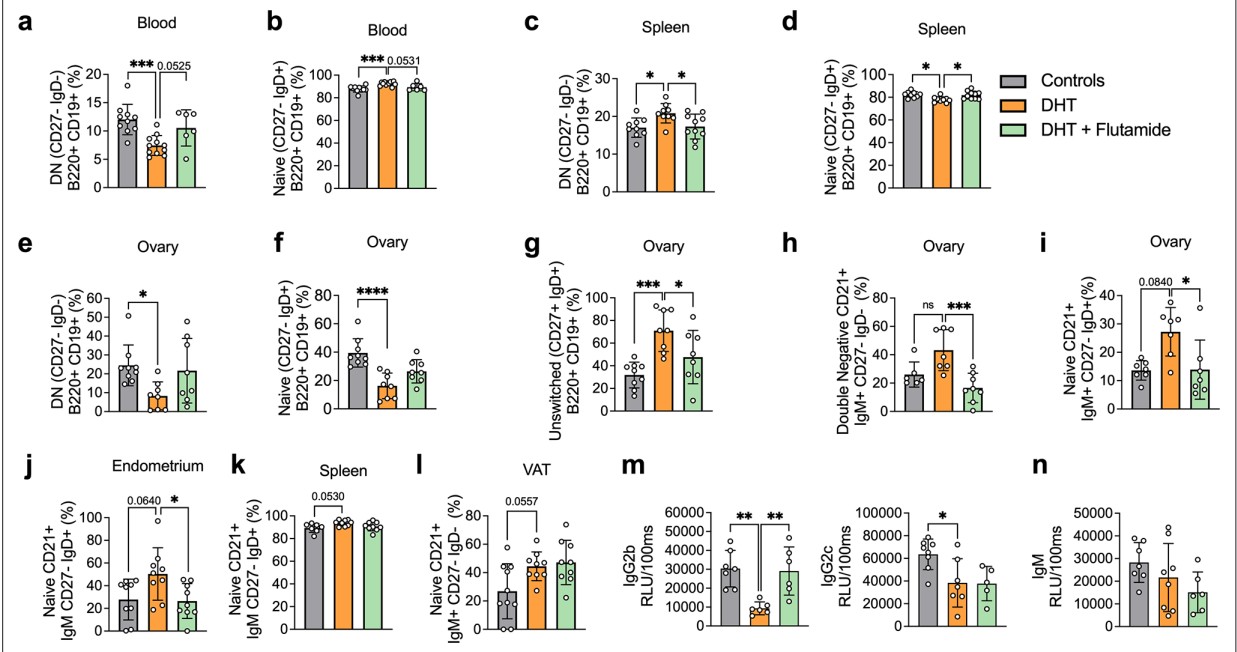

**Figure 5.** Dihydrotestosterone (DHT)-induced polycystic ovary syndrome (PCOS)-like mouse model B cell frequencies. (**a**) Blood double-negative (DN) B cells in 13-week-old mice. (**b**) Blood naive B cells in 13-week-old mice. (**a, b**) 13-week-old WT control mice (n = 10), mice receiving DHT pellet implant (n = 11), and mice receiving DHT pellet and flutamide implant (n = 10). (**c**) Spleen DN B cells in 20-week-old mice. (**d**) Spleen naive B cells in 20-week-old mice. (**e**) Ovary DN B cells in 20-week-old mice. (**f**) Ovary naive B cells in 20-week-old mice. (**g**) Ovary unswitched B cells in 20-week-old mice. (**h**) Ovary DN CD21+ B cells in 20-week-old mice. (**i**) Ovary naive CD21+ B cells in 20-week-old mice. (**j**) Visceral adipose tissue (VAT) naive CD21+ B cells in 20-week-old mice. (**k**) Spleen naive CD21+ B cells in 16-week-old mice. (**l**) Endometrium naive CD21+ B cells in 20-week-old mice. (**m**) Circulating IgG titers in 20-week-old mice. (**n**) Circulating IgM titers in 20-week-old mice. (**c–n**) 16–20-week-old WT control mice (n = 10), mice receiving DHT pellet implant (n = 9), and mice receiving DHT pellet and flutamide implant (n = 10). All bars indicate means, circles represent individual mice. In the case of missing values due to lack of measurement, mice were excluded from the analysis report for that variable. One-way ANOVA for multiple-comparisons of normally distributed data, Kruskal–Wallis test for data that is not normally distributed. *$p < 0.05$, **$p < 0.01$, ***$p < 0.001$.

The online version of this article includes the following figure supplement(s) for figure 5:

**Figure supplement 1.** Circulating antibody titers in dihydrotestosterone (DHT)-induced (PCOS)-like mouse model.

flutamide (*Figure 5h*), an effect previously observed in the blood of mice receiving IgG from women with PCOS. A similar increase was noted among naïve B cells of DHT-exposed mice, with a trend suggesting increased proportions of CD19+-naïve B cells expressing CD21+ in the ovaries (*Figure 5i*) and endometrium (*Figure 5j*), as well as spleen (*Figure 5k*) and VAT (*Figure 5l*). These trends were reversed by co-treatment with flutamide in ovary and endometrium.

Collectively, these results point to an inflammatory activity ongoing in the DHT-exposed mice presenting a PCOS-like phenotype, with B cell alterations being a consequence of AR activation as proven by the preventive effect of flutamide co-treatment. There are noticeable differences within the single tissues, which require further investigations.

## DHT-induced PCOS-like mice show a distinct IgG profile

In addition to functions deriving from T cell interaction, B cells regulate immune function via antibody production. Given the altered titers of IgM in women with PCOS, circulating IgM levels, as well as IgG isotypes, were analyzed in the peripubertal PCOS-like mouse model. DHT-exposed mice, exhibiting elevated levels of circulating testosterone and androstenedione, display reduced levels of IgG2b and IgG2c isotypes (*Figure 5m*), while no significant differences in IgM levels could be seen (*Figure 5n*). No differences were found for IgG1 nor in IgG3 titers (*Figure 5—figure supplement 1*).

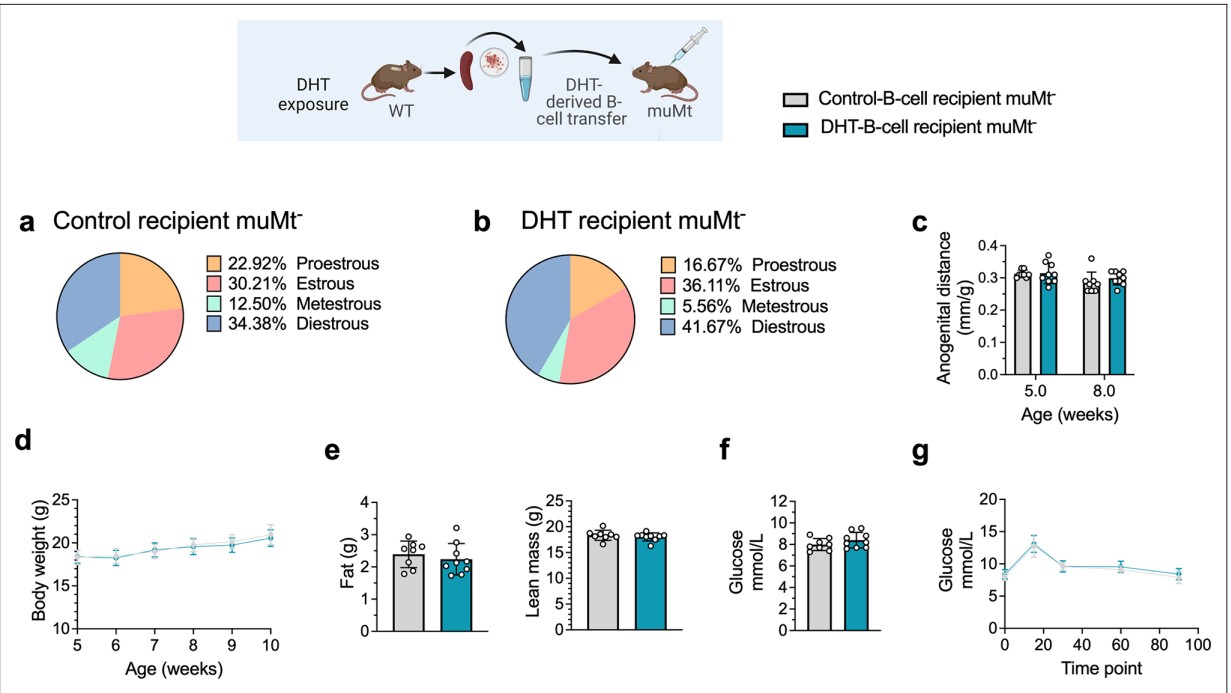

**Figure 6.** B cell transfer from dihydrotestosterone (DHT)-induced polycystic ovary syndrome (PCOS)-like mice into recipient muMt- B cell-deficient mice. (**a**) Estrous cycles in 13-week-old muMt- mice receiving control B cells. (**b**) Estrous cycles in 13-week-old muMt- mice receiving DHT exposed B cells. (**c**) Normalized anogenital distance in 13-week-old muMt- recipient mice. (**d**) Weekly body weight (BW) in 13-week-old muMt- recipient mice. (**e**) EchoMRI record of fat and lean body composition in 13-week-old muMt- recipient mice. (**f**) Fasting glucose levels. (**g**) Oral glucose tolerance test (OgTT) in 13-week-old muMt- recipient mice. (**a–g**) 13-week-old muMt- mice receiving control B cells (n = 8), 13-week-old muMt- mice receiving DHT-exposed B cells (n = 9). All bars indicate means, circles represent individual mice. In the case of missing values due to lack of measurement, mice were excluded from the analysis report for that variable. Unpaired Student's *t*-test for analysis of anogenital distance difference between groups, as well as EchoMRI results and fasting glucose; two-way ANOVA with Sidak's post hoc test for analysis of weekly BW recordings and blood glucose throughout the study; *p<0.05, **p<0.01, ***p<0.001.

## B cell transfer from DHT-induced PCOS-like mice into B cell-deficient mice does not induce a PCOS-like phenotype

To discern the role of B cells in the etiology of PCOS and the development of associated metabolic comorbidities, it was assessed whether transfer of B cells alone from DHT-exposed mice could induce a PCOS-like phenotype in B cell-deficient muMt- mice. Splenic B cells from DHT-exposed mice were transferred i.p. in to 6-week-old muMt- B[null] mice as they do not produce mature B cells due to the knockout of the mu heavy chain. It is important to note that they have, however, a fully functional T cell compartment. Control muMt- mice received an equal amount of splenic B cells deriving from a control donor group. Two weeks after transfer, the DHT-exposed B cells recipient muMt- mice failed to develop PCOS-like traits. Overall, B cell transfer did not affect the estrous cyclicity (*Figure 6a and b*), nor the anogenital distance (*Figure 6c*). Total body weight did not differ (*Figure 6d*), nor fat or lean mass (*Figure 6e*). Fasting glucose was not affected (*Figure 6f*) nor was glucose tolerance in oGTT testing (*Figure 6g*). The lack of a PCOS-like phenotype in a B cell reconstituted model with conserved T cell function was not explained by the presence of B cells alone to a hyperandrogenic environment and must therefore be driven by a peripheral mechanism that necessarily also affects function and properties of other immune cells.

## B cell deficiency does not protect from the induction of a PCOS-like phenotype by DHT exposure

To finally assess whether B cell deficiency provides a protective effect, 28-day/4-week-old muMt- mice, lacking mature B cells, were implanted with a silastic implant containing continuously releasing low-dose DHT. Control mice received a blank pellet. Four weeks after implantation, DHT-exposed B[null] muMt- developed a clear reproductive PCOS-like phenotype, exhibiting a disrupted estrous cycle

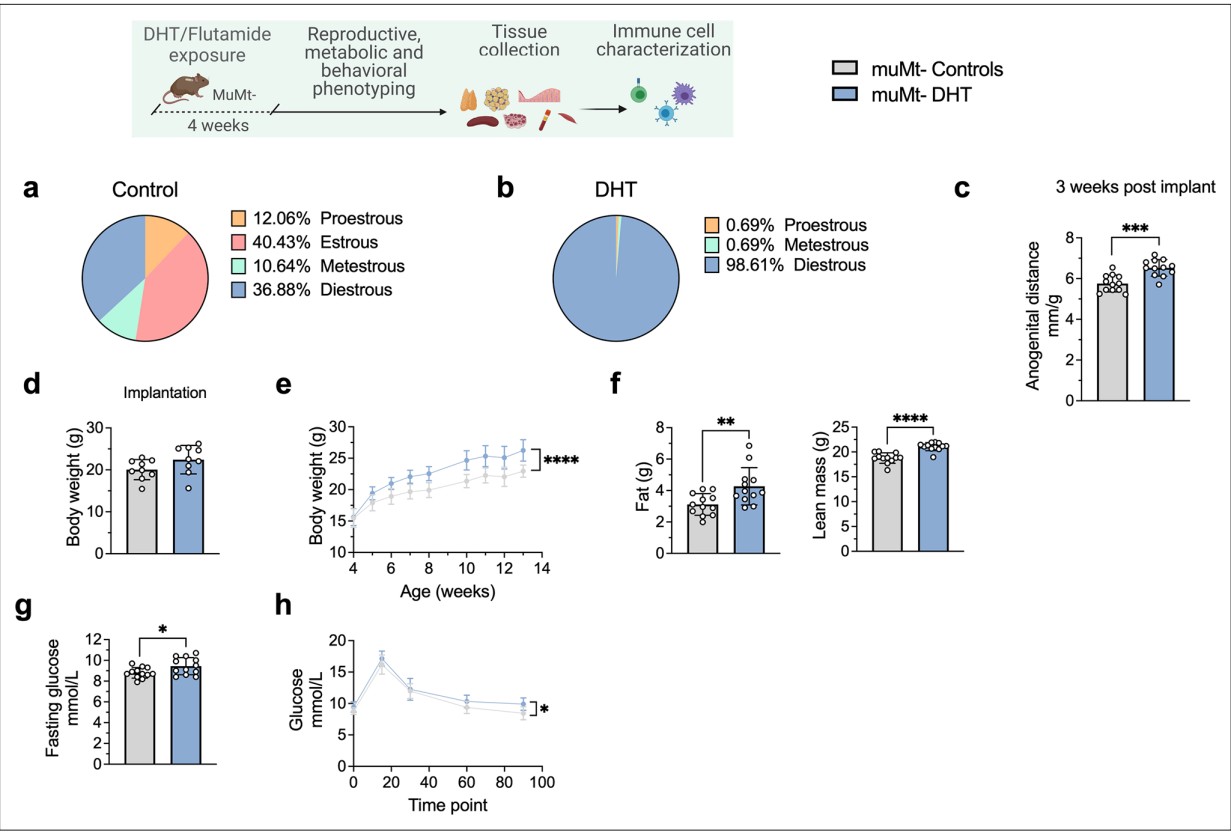

**Figure 7.** MuMt- dihydrotestosterone (DHT)-induced mouse model phenotypic study. (**a**) Estrous cycles in muMt- control mice. (**b**) Estrous cycles in muMt- mice receiving DHT pellet implant. (**c**) Normalized anogenital distance 3 wk post pellet implantation. (**d**) Body weight (BW) at pellet implantation. (**e**) Weekly BW recordings. (**f**) EchoMRI record of fat and lean body composition. (**g**) Fasting glucose levels. (**h**) Oral glucose test results. (**a–h**) muMt-control mice (n = 12), muMt- mice receiving DHT pellet implant (n = 12). All bars indicate means, circles represent individual mice. In the case of missing values due to lack of measurement, mice were excluded from the analysis report for that variable. Unpaired Student's *t*-test for analysis of anogenital distance difference between groups, as well as BW at implantation, fat mass, and fasting glucose; Mann–Whitney test for analysis of lean mass; two-way ANOVA with Sidak's post hoc test for analysis of weekly BW recordings and blood glucose throughout the study; *p<0.05, **p<0.01, ***p<0.001.

(*Figure 7a and b*), arrested in the diestrus phase, along with longer anogenital distance (*Figure 7c*). Furthermore, while no difference in body weight was noted amongst the groups at implantation (*Figure 7d*), DHT-exposed muMt⁻ mice gain higher body weight compared to controls already after 1 wk following implantation (*Figure 7e*), with increase both in total fat and lean mass (*Figure 7f*). When challenged to oGTT, DHT-exposed muMt⁻ mice exhibited impaired glucose homeostasis, with higher fasting glucose levels (*Figure 7g*) and higher blood glucose score 90 min after administration compared to control (*Figure 7h*).

These data firstly suggest that although affected in their function, B cells may not be central mediators of glucose metabolism impairment and reproductive dysfunction in PCOS. Furthermore, these mice develop a worsened outcome with increased adipose tissue that was unaltered in previous models, a risk factor of increased disease severity that is aggravated by obesity.

## Discussion

Here, we show a distinct link between hyperandrogenemia and abnormal B cell frequencies with circulating antibody titers in women with PCOS, potentially exerting effects systemically via pathogenic IgG antibodies inducing metabolic alterations leading to increased body weight as seen in mice following IgG transfer from these women. We further demonstrate how these highly regulated mechanisms are intrinsically AR activation-dependent. Most importantly, we show that B cells are not the central mediators of systemic inflammation or glucose metabolism impairment in PCOS as a lack of these lymphocytes does not protect from the induction of a PCOS-like phenotype following DHT

exposure. Rather the DHT-exposed muMt⁻ mice display an aggravated phenotype with higher fat mass accumulation.

In line with data combining increased BAFF levels and higher B cell frequencies in PCOS (*Xiao et al., 2019*), hyperandrogenic women with PCOS had a significant rearrangement of the B cell repertoire, resulting in higher frequencies of 'age-associated B cells' (ABCs) DN B memory cells that notably differ from naïve B cells for their lack of IgD. This occurred independently of the significant age difference noted between the two cohorts both randomly recruited as PCOS-affected women were notably younger than control participants. These are in fact age-associated tissue-based atypical memory B cells that represent a low-prevalence population, generally noted in the elderly, positively impacting immunosenescence (*Hao et al., 2011*). They have been however described in previous studies as autoantibody enriched, poised for plasma cell differentiation, and may overaccumulate prematurely in chronic infections, autoimmune diseases, and immunodeficiencies, thus playing a role in the regulation of humoral responses. However, based on B cells activation status and antibody-secreting potential, our findings did not support an ongoing proinflammatory activity driven by this specific heterogenous cluster as a major underlying mechanism in PCOS. Markers for B cell activation and T cell co-stimulatory capacity (CD86), as well as antibody secretion potential (CD38), did not differ from the DN memory B cells of our control population. Moreover, DN B memory cells deriving from women with PCOS were largely IgM-positive, resembling IgM-only cells. Previous studies have suggested this subset to be a major origin of switched-memory B cells (*Bagnara et al., 2015*). In humans, switched memory B cells have the propensity to differentiate into plasma cells upon reactivation (*Dogan et al., 2009*). These results may indicate that PCOS represents a state of prolonged trained immunity, susceptible to secondary triggers of inflammation. Consistent with this finding, by combining both markers IgD and CD27 in our study, we were able to distinguish a decrease in unswitched CD27⁺ IgD⁺ memory B cells accompanied by increased proportions of switched CD27⁺ IgD⁻ memory B cells in women with PCOS. Interestingly, a strong positive correlation was found between circulating IgM prevalence and androgens in women with PCOS, which was not affected by BMI. While increased adiposity is a determining factor in chronic inflammation, PCOS clearly triggers a condition-specific antigenic stimulus that is most likely independent of age, causing rearrangement of the humoral immune system.

There are data describing increased serum levels of autoantibodies (e.g., anti-histone and anti-double-stranded DNA antibodies) in PCOS (*Hefler-Frischmuth et al., 2010*). Additionally, a dose-dependent activation effect has been shown in vitro using PCOS-derived purified serum IgG antibodies on the gonadotropin-releasing hormone receptor (*Kem et al., 2020*) (GnRH). In the present study, the transfer of purified and concentrated serum IgG antibodies deriving from women affected with PCOS did increase body weight in recipient mice, although there was no biological variation in ovulatory cycles, a universal PCOS-like trait. A limitation of the study that should be noted is that the number of PCOS donors for IgG transfer was low with no significant difference in circulating androgen values, potentially affecting the biological effects resulting from the transfer. Based on the availability of purified serum, the number of recipient mice was also limited potentially leading to a loss of precision and statistical power in the case of missing values in the outcome. Nonetheless, interestingly we found a concomitant disruption of B cell frequencies and a sex hormone imbalance among estrogens, partially reflecting the status of PCOS donors. Recipient mice of human PCOS IgG had in fact increased circulating DN memory B cells, with tissue-specific higher proportions of unswitched memory cells in VAT and higher frequencies of switched memory B cells in the ovaries. These effects would suggest a regulated action by PCOS-derived IgG, altering energy metabolism that led to an immediate increase in body weight while concomitantly inducing a shift in both circulating and tissue-resident B cells toward altered cellular functionality. A metabolic shift could also further lead to a reprogramming of B cells toward a phenotype with a higher pro-inflammatory propensity as suggested by a recent study where peripheral B cells from women with PCOS were shown to have an increased capacity to produce TNF-α, which is attenuated by metformin treatment (*Xiao et al., 2022*). However, no variations were noted in the WT IgG-recipient mice metabolic phenotype as glucose metabolism was not affected. This would be in line with the observations of *Winer et al., 2011* already suggesting that preceding sterile inflammatory stimuli such as diet-induced pre-conditioning or induction of target autoantigens is required for proinflammatory IgG to have a critical role in rapid local and systemic metabolic changes. Such a stimulus affecting B cells frequencies may derive from direct or indirect testosterone/androgen receptor activation within the specific hyperandrogenic hormonal environment of PCOS.

Yet, although directly exposed to DHT before the onset of puberty, circulating B cell frequencies in the blood of peripubertal DHT-induced PCOS-like mouse model did not entirely reflect the distribution as seen in hyperandrogenic women. Additionally, we found a distinct profile of immunoglobulins in the peripheral blood of the peripubertal PCOS mouse model, with reduced levels of proinflammatory IgG2b and IgG2c titers. The increased concentration of circulating IgM in PCOS women was also not replicated. This raises the outstanding question regarding the influence of developmental origin and to what extent, if at all, does timing of androgen exposure alter B cell effector functionality in PCOS. Sensitivity of B cells to the influence of androgens is primarily during development as both pro- and pre-B cells within bone marrow express the AR, as do hematopoietic stem cells (HSCs) (*Gubbels Bupp and Jorgensen, 2018*). Indeed, during fetal and early postnatal origin unique innate-like lymphocytes, such as Vg3+ dendritic epidermal T cells as well as so-called B-1a cells, and a fraction of innate-like marginal zone B cells emerge that are not equally generated by adult bone marrow hematopoietic stem and progenitor cells (*Bendelac et al., 2001*; *Carey et al., 2008*; *Carvalho et al., 2001*; *Yoshimoto et al., 2011*). These cell types are characterized by distinct common features extending to both adaptive and innate immunity including their semi-invariant, often self-reactive antigen receptors, rapid responses to antigen stimulation and tissue resident localization (*Bendelac et al., 2001*). Hence, the variations of observed IgM titers in PCOS women may be due to different B-1a cells frequencies, responsible for spontaneous testosterone-independent secretion of natural IgM antibodies (NAbs), which represent a large extent of circulating serum IgM (*Boes, 2000*) that act with homeostatic housekeeping functions to multiple inflammatory reactions (*Lutz et al., 2009*). Although this remains purely speculative, as to whether this may be via immune activation related to the disease or a result of the disease driving the production of NAbs of IgM, we have previously shown that prenatal androgen exposure has severe effects on the health of offspring across generations, that is, transgenerational transmission (*Risal et al., 2019*), and alterations of IgM titers in lupus-prone (MRL/lpr) mice does induce a more severe autoimmunity (*Boes, 2000*). Furthermore, while the focus of our findings was on the B cell lineage, early androgen exposure in ontogeny may have a similar impact on T cells and other innate lymphoid cells, with significant implications for the development of PCOS pathology. However, we also know that regulation of B cell number by testosterone/AR may occur also in mature B cells lacking AR independently of developmental effects as demonstrated in spleen from short-term castration of adult mice (*De Gendt et al., 2004*). Within the context of sexual dimorphism in immunity and the effects of testosterone on B cells, previous research has focused on human male data or conditional AR knockout (ARKO) and castrated male mice. Both global ARKO models as well as castration cause a systemic alteration, which does not allow to distinguish in these studies if changes in the distribution of lymphocytes are due to a direct action of androgens on immature B cells expressing the AR or from secondary effects of surrounding tissue (*Lai et al., 2012*).

A recent study has demonstrated how endogenous testosterone regulates indirectly mature splenic B cell number in a BAFF-receptor dependent manner via testosterone-mediated increase in sympathetic nervous transmission regulating BAFF-producing fibroblastic reticular cells in male mice (*Wilhelmson et al., 2018*). There is, however, no data available regarding the concentration–response relation between BAFF levels, both local or systemic, and B cell numbers in mice. BAFF levels are overall tightly regulated and human studies suggest that serum BAFF levels within normal homeostatic ranges are inversely associated with peripheral B cell numbers (*Ferrer et al., 2012*; *Kreuzaler et al., 2012*). This has led to the notion that estradiol accelerates autoimmunity while testosterone has an inhibiting action (*Tedeschi et al., 2013*). However, the singular setting of hyperandrogenic PCOS, with increased circulating and tissue-specific levels of testosterone along with higher estrogen-to-progesterone ratios leading to anovulatory cycles, as well as high estrogen levels during prenatal life, may disrupt the development of the thymus and its function in maintaining immune tolerance and are suspected to heighten autoimmune response in PCOS (*Kowalczyk et al., 2017*). Indeed, serum levels of BAFF have been shown to be higher in women diagnosed with PCOS (*Xiao et al., 2019*). Our findings in the peripubertal DHT-induced PCOS mouse model showed metabolic alterations in glucose metabolism accompanied by variations among B cell populations, not only in blood-derived cells but also in immune cells seeding peripheral tissues. Trends of differential expression of complement receptor type 2 (CR2/CD21) on IgD$^+$-naïve cells in both metabolic, immune, and reproductive tissues may suggest an effect deriving from BAFF overexpression affecting transitional B cell maturation. BAFF is critical for the upregulation of expression of CD21, leading to greater proliferation and Ig

secretion potential (*Suryani et al., 2010*); however, there is no direct link with the development of an autoimmune state. These effects were prevented and overall reduced by concomitant treatment with flutamide, supporting the idea that any modulation of B cells is regulated via AR activation. Finally, the lack of phenotype entirely with no variations neither in metabolic nor reproductive parameters in B cell reconstituted muMt⁻ mice as well as in IgG-recipient T cell-deficient RAG1 KO⁻/⁻ mice further suggest that the effects of B cells deriving from a PCOS-like environment may be dependent on androgen exposure of other leukocytes to fully promote impairment of metabolic parameters.

The fact that B cell depletion has profound effects on glucose homoeostasis is well accepted. For example, rituximab, an anti-human CD20 mAb, used in the treatment of rheumatoid arthritis as well as B cell malignancies, can cause both hyperglycemia and severe hypoglycemia (*Hussain et al., 2006*). Depletion of B cells in mice with a CD20 mAb early in atherosclerotic disease has also shown therapeutic benefits in the abnormal glucose metabolism (*Ait-Oufella et al., 2010*). It is important to note that in these CD20 mAb therapies the beneficial effects were linked to reduced T cell activation. In the present study, the lack of B cells did not prevent the development of a PCOS-like phenotype in muMt⁻ mice exposed to DHT, with both reproductive and metabolic effects, such as higher deposits of adipose tissue. Based on these results, in accordance with the lack of phenotype in RAG1 KO⁻/⁻ mice following IgG transfer, identification of the precise role for T cell activation in PCOS warrants further investigation as well as of tissue-resident immune cells such as macrophages, potentially skewing to a pro-inflammatory phenotype, which may, in turn, be responsible for inflammatory cytokine production.

In conclusion, our study uncovers a previously unrecognized regulation via AR signaling, indirectly affecting B cell production of pathogenic IgG antibodies affecting energy metabolism. Moreover, our data raise a concern about uniquely identifying CD19⁺ B cells as a potential therapeutic target for PCOS as our findings do not support the notion that depletion of B cells is protective from developing PCOS-like traits. Concurrently increased levels of IgM may rather suggest a dual housekeeping function activity. Differences in the regulation of innate and adaptive immunity may be unique within the specific hormonal state of women with PCOS and should be investigated further.

# Materials and methods

**Key resources table**

| Reagent type (species) or resource | Designation | Source or reference | Identifiers | Additional information |
|---|---|---|---|---|
| Strain, strain background (*Mus musculus*) | C57BL/6JRj | Janvier Labs | | Female mice |
| Strain, strain background (*M. musculus*) | B6.129S7-Rag1, C57BL/6JRj | Jackson Laboratory | IMSR_JAX:002216 | Breeding pairs, homozygous for Rag1 |
| Strain, strain background (*M. musculus*) | B6.129S2-Ighm, C57BL/6JRj | Jackson Laboratory | IMSR_JAX:002288 | Breeding pairs, homozygous for Ighm |
| Antibody | Anti-human CD19-VioGreen, (clone REA675) (human monoclonal) | Miltenyi Biotec | AB_2726202 | 1:400 |
| Antibody | Anti-human anti-IgD-VioBlue (clone IgD26) (mouse monoclonal) | Miltenyi Biotec | AB_2659773 | 1:400 |
| Antibody | Anti-human CD27-APC (clone M-T271) (mouse monoclonal) | Miltenyi Biotec | AB_1036201 | 1:400 |
| Antibody | Anti-human CD86-PE-Vio770 (clone FM95) (mouse monoclonal) | Miltenyi Biotec | AB_275113 | 1:400 |
| Antibody | Anti-human CD38-FITC (clone IB6) (mouse monoclonal) | Miltenyi Biotec | AB_615091 | 1:400 |
| Antibody | Anti-human anti-IgM-PE (clone PJ2-22H3) (mouse monoclonal) | Miltenyi Biotec | AB_1036088 | 1:400 |
| Antibody | Alkaline phosphatase (AP)-labeled (goat polyclonal) anti-human IgM | μ-chain specific; Sigma-Aldrich | n/a | 1:50,000 in TBS BSA |

*Continued on next page*

*Continued*

| Reagent type (species) or resource | Designation | Source or reference | Identifiers | Additional information |
|---|---|---|---|---|
| Antibody | Alkaline phosphatase (AP)-labeled (goat polyclonal) anti-human IgG | γ-chain specific; Sigma-Aldrich | n/a | 1:50,000 in TBS BSA |
| Antibody | Alkaline phosphatase (AP)-labeled (goat polyclonal) anti-human IgA | α-chain specific; Sigma-Aldrich | n/a | 1:50,000 in TBS BSA |
| Antibody | Anti-mouse IgD-Pacific Blue (clone 11–26c.2a) (rat monoclonal) | BioLegend | AB_1937245 ( Cat#. 405711); AB_1937244 (Cat# 405712) | 1:400 |
| Antibody | Anti-mouse CD19-BV480 (clone 1D3) (rat monoclonal) | BD Biosciences | AB_2739509 | 1:400 |
| Antibody | Anti-mouse CD19-PE/Cyanine7 (clone 6D5) (rat monoclonal) | BioLegend | AB_313654 (Cat# 115519); AB_313655 (Cat# 115520) | 1:400 |
| Antibody | Anti-mouse CD45R/B220-FITC (clone RA3-6B2) (rat monoclonal) | BD Biosciences | AB_394618 | 1:400 |
| Antibody | Anti-mouse CD21/CD35-PE-CF594 (clone 7G6) (rat monoclonal) | BD Biosciences | AB_2738511 | 1:400 |
| Antibody | Anti-mouse CD138-PE/Cyanine7 Syndecan-1 (clone 281-2) (rat monoclonal) | BioLegend | AB_2562197 (Cat# 142513); AB_2562198 (Cat# 142514) | 1:400 |
| Antibody | Anti-mouse CD27-APC (clone LG.3A10) (Armenian hamster monoclonal) | BD Biosciences | AB_1727455 | 1:400 |
| Antibody | Anti-mouse IgM-APC/Cyanine7 (clone RMM-1) (rat monoclonal) | BioLegend | AB_10690815 (Cat# 406515); AB_10660305 (Cat# 406516) | 1:400 |
| Antibody | CD86-BV510 (clone GL1) (rat monoclonal) | BD Biosciences | Cat# 563077; RRID:AB_2737991 | 1:400 |
| Antibody | Anti-mouse IgM (μ-chain specific) (goat polyclonal M8644) | Sigma | MFCD00145913 | 2 µg/mL |
| Antibody | Anti-mouse IgG1 RMG1-1 (rat monoclonal) | BioLegend | AB_315060 (Cat# 406601); AB_315061 (Cat# 406602) | 2 µg/mL |
| Antibody | Anti-mouse IgG2b (clone R9-91) (rat monoclonal) | BD Biosciences | AB_394834 | 3 µg/mL |
| Antibody | Anti-mouse IgG2c (STAR135) (goat polyclonal) | Bio-Rad | AB_1102666 | 1 µg/mL |
| Antibody | Anti-mouse IgG3 (clone R2-38) (rat monoclonal) | BD Biosciences | AB_394841 | 4 µg/mL |
| Antibody | Anti-mouse IgA (clone C10-3) (rat monoclonal) | BD Biosciences | AB_396541 | 3 µg/mL |
| Antibody | Anti-mouse IgG1 (clone A85-1) (rat monoclonal) | BD Biosciences | AB_393553 | |
| Antibody | Anti-mouse IgG2b (clone R12-3) (rat monoclonal) | BD Biosciences | | |
| Antibody | Anti-mouse AffiniPure anti-mouse IgG, Fcγ subclass 2c specific, (goat polyclonal) | Jackson | JIR 115-065-208 | |
| Antibody | Anti-mouse IgG3 (clone R40-82) (rat monoclonal) | BD Biosciences | | |
| Antibody | Anti-mouse IgA (clone C10-1) (rat monoclonal) | BD Biosciences | | |
| Commercial assay or kit | HiTrap Protein G HP purification column | Bio-Sciences AB | GE17-0404-01 | |
| Commercial assay or kit | Amicon Ultra-15 Centrifugal Filters | Merck Millipore | 30 kDa MWCO | |

*Continued on next page*

*Continued*

| Reagent type (species) or resource | Designation | Source or reference | Identifiers | Additional information |
|---|---|---|---|---|
| Commercial assay or kit | Automated chemiluminescence immunoassay | | ADVIA Centaur XP | |
| Commercial assay or kit | ELISA kit | | Crystal Chem | |
| Chemical compound, drug | Lumi-Phos | | Lumigen | 33% solution in water |

## Human case–control explanatory study cohort

For our first aim, to evaluate B cell frequencies and particularly the distribution of DN B cells among women with PCOS, from September 2019 to March 2021, 42 women, all Caucasian ethnicity, were screened for a PCOS diagnosis at the Medical University Clinics in Graz (Austria) for either of the two main PCOS hyperandrogenic phenotypes. Phenotype A: clinical hyperandrogenemia, oligo- anovulation, and PCOM; or phenotype B: clinical hyperandrogenemia and oligo- anovulation. Clinical hyperandrogenism was assessed using the modified Ferriman–Gallwey (FG) score, with a self-reported score of 8 or higher indicating hirsutism (*Yildiz et al., 2010*). For total testosterone, a cut-off of 0.6 ng/mL (2.1 nmol/L) was used based on previously published data from a representative population sample (*Lindheim et al., 2017*). Oligo-/anovulation was defined as menstrual cycles with a duration >35 d or the absence of menstruation for three or more consecutive months. PCOM, diagnosed by a gynecological ultrasound, was assessed based on medical history. Thyroid disorder, congenital adrenal hyperplasia, Cushing's syndrome, hyperprolactinemia, androgen-secreting tumors, and pregnancy were excluded by laboratory measurements of thyroid-stimulating hormone (TSH), 17-hydroxyprogesterone (17OH-P), cortisol, prolactin, pregnancy test, and clinical examination. Exclusion criteria considered multiple factors affecting participants' immune and hormonal profiles, such as neoplastic, infectious, and autoimmune diseases as well as currently used hormonal contraceptives or immunomodulating drugs. The final analyses of fasting blood samples for B cell frequencies were performed from a group of 15 women with PCOS and 22 controls. Antibody variation titers were examined in the same cohort of women specifically in 15 PCOS and in 18 of the 22 controls. In the case of missing values due to lack of measurement, patients were excluded from the analysis report for that variable. All recruitment took place at the endocrinological Outpatient Clinic of the University Hospital Graz by routine doctors, and nurses involved in the project. All participants provided oral and written informed consent after a positive vote of the Ethics Committee of the Medical University Graz (EK 31-560 ex 18/19). The work here described has been carried out in accordance with The Code of Ethics of the World Medical Association (Declaration of Helsinki) for experiments involving humans. For the extraction of IgG, from February 2020 to October 2020, a second cohort of 10 women were randomly recruited to voluntarily participate in the study for transfer of purified antibodies. Seven women were diagnosed with PCOS, one did not fulfill the inclusion criteria and two decided to drop out, leaving four women with PCOS and four healthy donors. The diagnosis was conducted at the Medical University Clinics in Graz (Austria) according to the aforementioned criteria.

## Clinical examination, blood sampling, and biochemical measurements

Anthropometric measures included weight, height, waist circumference, and BMI, which was calculated as weight (kg)/height (m) (*Gaberšček et al., 2015*) and waist-to-hip circumference. Baseline fasting blood samples were drawn for each participant in serum, EDTA, and lithium heparin tubes. Hormonal levels were assessed in fasting serum samples: total and free testosterone, androstenedione, and progesterone were measured by liquid chromatography–tandem mass spectrometry as described elsewhere (*Lindheim et al., 2017*); sex hormone-binding globulin (SHBG), anti-Müllerian hormone (AMH), and insulin were measured by automated chemiluminescence immunoassay (ADVIA Centaur XP, Roche, Rotkreuz, Switzerland); serum luteinizing hormone (LH) and follicle-stimulating hormone (FSH) were measured by enzyme-linked immunosorbent assay (ELISA, DIAsource Immunoassay, Belgium); plasma total cholesterol, high-density lipoprotein (HDL) cholesterol, triglycerides, and glucose were measured by automated enzymatic colorimetric assay (Cobas, Roche, Germany). The area under the curve (AUC) for glucose and insulin was calculated from the oGTT using the trapezoidal

method. Serum SHBG and testosterone were used to calculate the free androgen index as serum testosterone/SHBG ×100.

## Chemiluminescent ELISA

Chemiluminescent ELISA of human samples was performed as described elsewhere (*Hendrikx et al., 2016*) for total IgM, IgG, and IgA. In brief, purified anti-human IgM, IgG, and IgA (BD Pharmingen, San Jose, CA) at concentrations of 5 µg/mL in 50 µL phosphate-buffered saline (PBS)-EDTA were added to each well of a 96-well white, round-bottom microtitration plate (MicrofluorII round-bottom; Thermo, Rochester, NY) and incubated overnight at 4°C. After washing and blocking with Tris-buffered saline (TBS) with or without EDTA (pH 7.4, containing 1% bovine serum albumin [BSA], 30 min at room temperature [RT]), the plate was incubated with plasma samples in their respective dilutions in 1% BSA in TBS with EDTA (pH 7.4) for 2 hr at RT or overnight at 4°C. Alkaline phosphatase (AP)-labeled goat anti-human IgM (µ-chain specific; Sigma-Aldrich, Vienna, Austria; 1:50,000 in TBS BSA), AP-labeled goat anti-human IgG (γ-chain specific; Sigma-Aldrich, Vienna, Austria; 1:50,000 in TBS BSA), and AP-labeled goat anti-human IgA (α-chain specific; Sigma-Aldrich, Vienna, Austria; 1:50,000 in TBS BSA) were used for detection. AP-conjugated secondary reagents were detected using Lumi-Phos (Lumigen, Southfield, MI; 33% solution in water) and a Synergy 2 Luminometer (BioTek, Winooski, VT). Washing steps were performed on an ELx405 Select Deep Well Microplate Washer (BioTek) with PBS or PBS-EDTA. Internal controls were included on each microtiter plate to detect potential variations between microtiter plates. The intra-assay coefficients of variation for all assays were 5–15%.

## Lymphocyte phenotyping of human samples

Blood samples from the baseline visit were processed within 4 hr for analysis by flow cytometry as previously described (*Schulz et al., 2021*). Briefly, for B-cell phenotyping, PBMCs were isolated from lithium heparin whole blood by Ficoll gradient density centrifugation. One million PBMCs were incubated with the following antibodies: CD19-VioGreen (clone REA675), anti-IgD-VioBlue (clone IgD26), CD27-APC (clone M-T271), CD86-PE-Vio770 (clone FM95), CD38-FITC (clone IB6), and anti-IgM-PE (clone PJ2-22H3, all purchased from Miltenyi Biotec, Bergisch Gladbach, Germany). Samples were measured using a FACSLyric flow cytometer (BD Biosciences, Franklin Lakes, NJ). Data were analyzed using the FACSSuite (BD Biosciences).

## Animals and study design

All mice experiments were carried out in compliance with the ARRIVE guidelines in accordance with the U.K. Animals (Scientific Procedures) Act, 1986, and associated guidelines, EU Directive 2010/63/EU for animal experiments. All animal experiments were approved by the Stockholm Ethical Committee for animal research (20485-2020) in accordance with the Swedish Board of Agriculture's regulations and recommendations (SJVFS 2019:9) and controlled by Comparative Medicine Biomedicum at the Karolinska Institutet in Stockholm, Sweden. Mice were maintained under a 12 hr light/dark cycle and in a temperature-controlled room with ad libitum access to water and a diet. All mice were on female on C57BL/6J background. For the transfer of human IgG 24 five-week-old female C57BL/6JRj mice were obtained from Janvier Labs. Rag1 KO$^{-/-}$ were generated by breeding 10 male and 10 female B6.129S7-Rag1 (homozygous for Rag1) breeding pairs from Jackson Laboratory. For immune characterization of the peripubertal DHT-induced PCOS mice, 30 three-week-old female C57BL/6JRj mice were obtained from Janvier Labs and left to acclimatize for 1 wk. For the B cell reconstitution, 10 three-week-old female C57BL/6JRj mice were obtained from Janvier Labs to develop the peripubertal DHT-induced PCOS model. This peripubertal DHT-induced PCOS mouse model was developed by implanting a 5 mm silastic implant containing 2.0–2,5 mg of continuously releasing DHT according to previously published protocol (*Xue et al., 2018*), which was implanted subcutaneously in the neck region of 28–29-day-old C57BL/6JRj female mice. Surgery was performed under light anesthesia with isoflurane. Control mice were implanted with an empty, blank implant. To investigate androgen receptor activation, a third group received, in addition to the DHT implant, a 4.5 mm continuously releasing pellet containing 25 mg of flutamide (releasing time 90 d, Innovative Research of America, Cat# NA-152), an androgen receptor antagonist. Mice were randomly allocated to one of these three groups: control, DHT, and DHT-flutamide. A PCOS-like phenotype was fully developed after 3 weeks of exposure. MuMt$^-$ mutant mice were generated from 10 male and 10 female

B6.129S2-Ighm (homozygous for Ighm) breeding pairs from Jackson Laboratory. No mice received further monthly implants.

## Purification and transfer of IgG

IgG from human sera was purified utilizing a HiTrap Protein G HP purification column (Bio-Sciences AB) according to the vendor's instructions. Briefly, samples were centrifuged at 3000 RCF for 5 min at 4°C and supernatant was diluted 5× with binding buffer. The final elution containing IgG was dialyzed overnight at 4°C against endotoxin-free PBS and further filtered to obtain sterile antibody solution. IgG concentration in each sample was measured by QUBIT (Thermo Scientific) according to the vendor's instructions and stored at –20°C. Samples from the serum of PCOS-affected women cohort or serum of the control group were separately pooled. The day before injection, samples were filtered and concentrated using Amicon Ultra-15 Centrifugal Filters (30 kDa MWCO – 15 mL sample volume) according to the vendor's instructions (Merck Millipore). Briefly, samples were thawed and kept at 4°C the night before concentration; after filtering samples through sterile 0.22 μm syringe filter, desired concentration was obtained by spinning multiple times at 1000 × $g$/4°C until reaching final concentration of 4 mg/mL of IgG antibody in a total volume of 450 μL at injection days 1 and 3, and 365 μL at injection day 10 of endotoxin-free PBS. Final IgG concentration was measured once again by QUBIT (Thermo Scientific). 7-week-old female C57BL/6JRj mice, randomly divided into two study groups of six mice each, received purified human IgG (>98% pure) via i.p. injection in endotoxin-free PBS on days 1, 3, and 10. To assess the role of T cells mediating the response to IgG, the same procedure was repeated utilizing age-matched in-house bred mutant Rag1 KO[-/-] mice.

## Assessment of reproductive phenotype

In all groups, anogenital distance, a biomarker for androgen exposure, was measured at baseline and at sacrifice. For the transfer of human IgG, anogenital distance was measured 1 wk after first i.p. injection in both WT and RAG1 KO[-/-] mice. For immune characterization, anogenital distance was measured 3 wk after DHT/flutamide implantation. For B cell reconstitution, anogenital distance in reconstituted muMt- mice was measured 2 wk after reconstitution. Estrous cyclicity was assessed by daily vaginal smear for 12 consecutive days (three ovulatory cycles).

## Assessment of metabolic phenotype

Body weight development was recorded weekly. Body composition was assessed by magnetic resonance imaging (EchoMRI-100 system, Houston, TX) to measure total fat and lean mass in conscious mice. Glucose metabolism was measured by oGTT after a 5 hr fast. Mice received 2 mg per gram body weight of D-glucose (20% glucose in 0.9% NaCl) administrated by orally by gavage. Blood glucose was measured at baseline and at 15, 30, 60, and 90 min following glucose administration (Free Style Precision). Blood was collected in EDTA-coated capillary tubes at baseline and 15 min for insulin measurement by tail bleeding. Plasma separation is obtained by spinning the samples at 2000 × $g$ for 10 min at 4°C and stored at – 20°C. Based on the study design for individual project objectives, for the transfer of human IgG as well as B cell reconstitution, mice were first assessed for oGTT when the expected effects from the transfer on glucose metabolism were most likely at their peak, followed by EchoMRI evaluation. For project characterization of DHT-induced PCOS-like mouse model as well as the characterization of androgen-exposed muMt- mouse model, mice were first screened through EchoMRI to measure total fat and lean mass and then subjected to an oGTT evaluation.

## Biochemical assessment of insulin and sex steroids in mice

Plasma insulin from oGTT was analyzed by an ELISA kit (Crystal Chem). Testosterone, androstenedione, estradiol, estrone, and progesterone were measured in serum using a high-sensitivity liquid chromatography–tandem mass spectrometry assay as previously described (*Ohlsson et al., 2022*).

## Tissue collection and cell isolation

For the transfer of human IgG, C57BL/6JRj WT mice were sacrificed at 13–14 weeks of age. Rag1 KO[-/-] receiving human IgG were sacrificed at 16–17 weeks of age. For immune characterization of the peripubertal DHT-induced PCOS model, two independent experiments were conducted to evaluate separate timepoints: a first cohort of C57BL/6JRj mice were sacrificed at 20–22 weeks of age, while in

a following assessment DHT-exposed C57BL/6JRj mice were sacrificed at 13–14 weeks of age. For the reconstitution of B^null muMt^- mice with splenic B cells following DHT exposure, a cohort of C57BL/6JRj mice were sacrificed at 8 weeks of age, 4 weeks after DHT implant, for the retrieval of spleen B cells. The B cell reconstituted muMt^- mice were sacrificed at 11–12 weeks of age. For characterization of DHT-exposed MuMt^- model, mice were sacrificed at 13–14 weeks of age. All mice were sacrificed based on their ovulatory cycle stage in metestrus or diestrus, assessed by vaginal smears less than 2 hr prior sacrifice. Mice were fasted for 2 hr and anesthetized with isoflurane (Isoflo vet, Orion Pharma Animal Health). Blood was drawn by cardiac puncture using a 21G needle; an aliquot of 150 μL was directly transferred to EDTA-coated tube, and placed on ice for FACS analysis. The remaining amount of blood was transferred to microvette capillary tubes (Sarstedt) for serum separation. After dissection, spleen and lymph nodes were kept on ice in PBS without $Ca^{2+}$ and $Mg^{2+}$ (DPBS). Ovaries, endometrium, and VAT tissues were maintained in RPMI containing 2% FBS on ice for cell isolation. For analysis of sex steroid, serum in aliquots of 250 μL was separated by centrifugation at 5000 × $g$ for 10 min at 4°C.

## Comprehensive B lymphocyte phenotyping of mice tissues

To obtain single cells, spleen and inguinal and retroperitoneal lymph nodes were directly passed through a nylon wool sieve (100 μm cell strainer). After centrifugation at 300 RCF at 4°C for 5 min, erythrocytes (in spleen) were hemolyzed in 1 mL red blood cell lysis buffer (RBC lyse buffer; 0.16 M $NH_4Cl$, 0.13 mM EDTA, and 12 mM $NaHCO_3$ in $H_2O$), followed by a wash in 2 mL of FACS buffer (×2 the volume of RBC lysis). After a second centrifugation at 300 RCF at 4°C for 5 min, cells were resuspended in flow cytometry buffer (2% fetal bovine serum and 2 mM EDTA in PBS). Ovarian and uterus tissues were transferred into a 1 mL and 3 mL digestive mix (1 mg/mL collagenase type I from 210 U/mg, 0.8 U DNase I, RPMI, 2% FBS), respectively, minced by fine scissors and digested by gentle shaking for 15 and 20 min, respectively, at 37°C. To inactivate the enzymatic activity, 2 mL and 6 mL, respectively, of cold flow cytometry buffer was added to ovaries and uterus and placed on ice before grinding tissues through a 100 μm cell strainer. Samples were spun at 1000 × $g$ for 7 min at 4°C and resuspended in flow cytometry buffer. VAT was minced by fine scissors in 5 mL digestive buffer based on RMPI containing 2% of FBS and 1 mg/mL collagenase type IV (type D, 0.15 U/mg) and digested by gentle shaking for 20–25 min at 37°C. To inactivate the enzymatic activity, 10 mL of cold flow cytometry buffer was added and placed on ice before filtering suspensions through 100 μm filter and further spinning at 500 × $g$ for 5 min. The resuspended cell pellet was left for 30 s at RT in 500 μL RBC lyse buffer, then washed in 1 mL of FACS buffer (×2 the volume of RBC lysis) and centrifuged at 500 × $g$ for 5 min at 4°C. Blood volume of approximately 120 μL was placed twice in 1 mL of RT RBC lysis buffer (for an approximate proportion of 1:10) for 5 and 2 min, respectively, each time diluted in 2 mL of flow cytometry buffer and spun at 380 × $g$ for 5 min at 4°C. All tissue-deriving cells were plated on 96-well round-bottom plates and stained (Sarstedt, 83.3925.500) with FC-blocking antibody surface antigen staining (CD16/32, clone 2.4G2, BD Biosciences) diluted 1:100 in flow cytometry buffer, followed by incubation with the following antibodies: IgD-Pacific Blue (clone 11-26c.2a, BioLegend), CD19-BV480 or PE/Cyanine7 (clone 1D3, BD Biosciences, or clone 6D5, BioLegend, respectively), CD45R/B220-FITC (clone RA3-6B2, BD Biosciences), CD21/CD35-PE-CF594 (clone 7G6, BD Biosciences), CD138-PE/Cyanine7 (Syndecan-1, clone 281-2, BioLegend), CD27-APC (clone LG.3A10, BD Biosciences), IgM-APC/Cyanine7 (clone RMM-1, BioLegend), and CD86-BV510 (clone GL1, BD Biosciences). Samples were measured using a FACS Canto II flow cytometer (BD Biosciences). Data were analyzed using FlowJo (BD Biosciences).

## Total antibody quantification in plasma by ELISA

Chemiluminescent ELISA was performed as described elsewhere (*Tsiantoulas et al., 2021*). Total IgM, IgG1, IgG2b, IgG2c, IgG3, and IgA antibodies in plasma were measured by ELISA. In brief, 96-well white round-bottomed MicroFluor microtiter plates (Thermo Lab Systems) or immunoGrade, 96-well, PS Standard plates (781724; Brand) were coated with an anti-mouse IgM (Sigma; M8644; at 2 μg/mL), anti-mouse IgG1 (BioLegend; RMG1-1; at 2 μg/mL), anti-mouse IgG2b (BD Biosciences; R9-91; at 3 μg/mL), anti-mouse IgG2c (STAR135; at 1 μg/mL), anti-mouse IgG3 (BD Biosciences; R2-38; at 4 μg/mL), or anti-mouse IgA (BD Biosciences; C10-3; at 3 μg/mL) in PBS overnight and then washed three times with PBS and blocked with Tris-buffered saline containing 1% BSA (TBS/

BSA) for 1 hr at RT. Then wells were washed with either PBS (plates for IgM, IgG2b, and IgG2c) or PBS supplemented with 0.05% Tween (plates for IgG1, IgG3, and IgA), and diluted mouse plasma was added in TBS/BSA to the wells and incubated overnight at 4°C. Plates were washed, and bound Igs were detected with an anti-mouse IgM antibody conjugated to alkaline phosphatase (Sigma; A9688), the biotinylated forms of anti-mouse IgG1 (BD Biosciences; A85-1) or anti-mouse IgG2b (BD Biosciences; R12-3), anti-mouse IgG2c (JIR 115-065-208), anti-mouse IgG3 (BD Biosciences; R40-82), or anti-mouse IgA (BD Biosciences; C10-1). Wells were washed again as before and neutravidin conjugated to alkaline phosphatase was added where appropriate. Then, wells were washed again as before and rinsed once with distilled water, and 25 µL of a 30% LumiPhos Plus solution in dH$_2$O (Lumigen Inc) was added. After 75 min, the light emission was measured with a Synergy 2 luminometer (BioTek) and expressed as RLU per 100 ms.

## Statistics

For statistical evaluation, Prism (version 9; GraphPad Software) and SPSS (version 28.0; SPSS) were used. All continuous data were screened for normality by Shapiro–Wilk test and equality of variance. Normally distributed data were compared using unpaired Student's $t$-tests, and when not normally distributed differences between groups were compared using Mann–Whitney $U$-test. Differences between more than two groups were determined by ANOVA followed by Tukey's post hoc test. Differences were considered statistically significant at $p < 0.05$. One patient or one animal was considered a biological replicate. In the case of missing values, patients or animals were excluded from the analysis for that variable.

For the first cohort, the human monocentric case–control explanatory study based on B cell frequencies, the sample size was calculated taking into account the distribution associated with specific PCOS phenotypes A and B. More than half of PCOS patients identified within the clinical setting demonstrate phenotype A, whereas the other three phenotypes (i.e., B, C, and D) have almost equal prevalence (*Lizneva et al., 2016*), added to the observations that the presence of hyperandrogenism (*Ehrmann et al., 2006*), BMI (*Ehrmann et al., 2006*), and degree of menstrual irregularity (*Brower et al., 2013*), while no ovarian morphology (*Legro et al., 2005*), may be considered independent predictors of metabolic dysfunction. Finally, with the aim to evaluate the effects of double negative autoreactive B cells and assuming a predicted variation of 10% among total CD19$^+$ B cell populations in PCOS patients based on previously reported assessments (*Xiao et al., 2019*), we aimed for a total sample of 40 subjects, 20 PCOS, and 20 controls, to achieve 90% power to detect differences among the means versus the alternative of equal means using an $F$ test with a 0.05 significance level. The size of the variation in the means is represented by the effect size f = σm/σ, which is 0.31. Sample size was generated using PASS 15.0.6.

For the second study cohort, for the transfer of purified IgG, a predefined sample size was not assigned.

For animal studies, no statistical methods were used to predetermine sample size, based on previous reported assessments (*Risal et al., 2019*). Animals were allocated to experimental groups arbitrarily without formal randomization. Investigators were not formally blinded to group allocation during the experiment.

## Acknowledgements

This work was supported by grants from the Swedish Medical Research Council: project no. 2018-02435 and 2022-00550 (ESV); Novo Nordisk Foundation: Distinguished Investigator Grant – Endocrinology and Metabolism, NNF22OC0072904 and NNF19OC0056647 (ESV); Diabetes Foundation: DIA2021-633 and DIA2022-708 (ESV); Strategic Research Program in Diabetes at the Karolinska Institutet (ESV); Karolinska Institutet KID funding: 2020-00990 (ESV); Regional Agreement on Medical Training and Clinical Research between the Stockholm County Council and the Karolinska Institutet: 20190079 (ESV); EMBO Scientific Exchange Grants 2021: STF 8938 (AA); European Research Council (ERC) under the European Union's Horizon 2020 research and innovation program under the grant agreement no. 866075 (CIS); Knut and Alice Wallenberg Foundation no. 018.0161 (CIS); Austrian Science Fund (FWF) project number W1241 (BOP).

# Additional information

## Funding

| Funder | Grant reference number | Author |
| --- | --- | --- |
| Vetenskapsrådet | 2018-02435 | Elisabet Stener-Victorin |
| Vetenskapsrådet | 2022-00550 | Elisabet Stener-Victorin |
| Novo Nordisk Fonden | NNF22OC0072904 | Elisabet Stener-Victorin |
| Novo Nordic Foundation | NNF19OC0056647 | Elisabet Stener-Victorin |
| Diabetesfonden | DIA2021-633 | Elisabet Stener-Victorin |
| Diabetesfonden | DIA2022-708 | Elisabet Stener-Victorin |
| Karolinska Institutet | Strategic Research Program in Diabetes | Elisabet Stener-Victorin |
| Karolinska Institutet | KID funding: 2020-00990 | Elisabet Stener-Victorin |
| Stockholm County Council | Regional Agreement on Medical Training and Clinical Research between the Stockholm County Council and the Karolinska Institutet: 20190079 | Elisabet Stener-Victorin |
| EMBO | Scientific Exchange Grants 2021 STF 8938 | Angelo Ascani |
| Horizon 2020 - Research and Innovation Framework Programme | 866075 | Camilla I Svensson |
| Knut and Alice Wallenberg Foundation | 018.0161 | Camilla I Svensson |
| Austrian Science Fund | W1241 | Barbara Obermayer-Pietsch |

The funders had no role in study design, data collection and interpretation, or the decision to submit the work for publication.

## Author contributions

Angelo Ascani, Conceptualization, Data curation, Formal analysis, Validation, Visualization, Methodology, Writing – original draft, Writing – review and editing; Sara Torstensson, Conceptualization, Data curation, Formal analysis, Validation, Investigation, Visualization, Methodology, Writing – original draft, Writing – review and editing; Sanjiv Risal, Gustaw Eriksson, Sabrina Teschl, Joana Menezes, Claes Ohlsson, Martin Helmut Stradner, Investigation, Writing – review and editing; Haojiang Lu, Investigation, Methodology, Writing – original draft, Writing – review and editing; Congru Li, Katalin Sandor, Investigation, Methodology, Writing – review and editing; Camilla I Svensson, Funding acquisition, Investigation, Methodology, Writing – review and editing; Mikael CI Karlsson, Data curation, Investigation, Methodology, Writing – review and editing; Barbara Obermayer-Pietsch, Conceptualization, Supervision, Funding acquisition, Investigation, Writing – original draft, Writing – review and editing; Elisabet Stener-Victorin, Conceptualization, Resources, Data curation, Supervision, Funding acquisition, Methodology, Writing – original draft, Project administration, Writing – review and editing

## Author ORCIDs

Sara Torstensson (ID) http://orcid.org/0000-0003-4389-2662
Gustaw Eriksson (ID) http://orcid.org/0000-0003-0120-9028
Katalin Sandor (ID) http://orcid.org/0000-0003-3228-6907
Barbara Obermayer-Pietsch (ID) https://orcid.org/0000-0003-3543-1807
Elisabet Stener-Victorin (ID) http://orcid.org/0000-0002-3424-1502

### Ethics

Participants provided oral and written informed consent after a positive vote of the Ethics committee of the Medical University Graz (EK 31-560 ex 18/19). The work here described has been carried out in accordance with The Code of Ethics of the World Medical Association (Declaration of Helsinki) for experiments involving humans.

All animal experiments were approved by the Stockholm Ethical Committee for animal research (20485-2020) in accordance with the Swedish Board of Agriculture's regulations and recommendations (SJVFS 2019:9) and controlled by Comparative Medicine Biomedicum at the Karolinska Institutet in Stockholm, Sweden.

### Decision letter and Author response

Decision letter https://doi.org/10.7554/eLife.86454.sa1
Author response https://doi.org/10.7554/eLife.86454.sa2

---

## Additional files

### Supplementary files
• MDAR checklist

### Data availability

All data generated or analysed during this study are included in the manuscript and supporting files Raw data can be found at Mendeley Data: https://doi.org/10.17632/tcc2mbmys4.1.

The following dataset was generated:

| Author(s) | Year | Dataset title | Dataset URL | Database and Identifier |
|---|---|---|---|---|
| Ascani A, Torstensson S, Risal S, Lu H, Eriksson G, Li C, Teschl S, Menezes J, Sandor K, Ohlsson C, Svensson C, Karlsson M, Stradner M, Obermayer-Pietsch B, Stener-Victorin E | 2023 | The role of B cells in immune cell activation in polycystic ovary syndrome | https://doi.org/10.17632/tcc2mbmys4.1 | Mendeley Data, 10.17632/tcc2mbmys4.1 |

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
