## [Editor Report]

This manuscript provides the fundamental findings regarding the alteration of B cell frequencies and functionality, caused by the activation of androgen receptor. Based on the compelling strength of evidence, the manuscript presents significant results improving the current understanding of PCOS and associated disorders. The manuscript will be of interest to medical biologists, gynaecologists, and endocrinologists.

---

## [Decision Letter]

**Decision letter after peer review:**

Thank you for submitting your article "The role of B cells in immune cell activation in polycystic ovary syndrome" for consideration by *eLife*. Your article has been reviewed by 2 peer reviewers, one of whom is a member of our Board of Reviewing Editors, and the evaluation has been overseen by Diane Harper as the Senior Editor. The following individual involved in the review of your submission has agreed to reveal their identity: Licy Yanes Cardozo (Reviewer #3).

Essential revisions:

Please, consider this list of essential issues that could help you to improve the clarity and presentation of your work:

1) Check the number of subjects at each stage of study and report it correctly in the tables and figures.

2) Explain the reasons for the use of T-test criteria for some comparisons presented in Tables S1-2.

3) Discuss the limitations of the study considering sources of potential bias.

*Reviewer #1 (Recommendations for the authors):*

The study is well-designed, the methods are convincing. Generally, results are based on the compelling strength of evidence, but there are some remarks regarding the data presentation.

The total number of screened women should be presented in the Material and Methods section (page 13), because n=15 is the final number of recruited women with PCOS.

It looks reasonable to present the number of observations in all the Figures.

Please, comment on the significant age difference between women from the PCOS and control groups, which may cause some biases.

At the same time, the use of the T-test when comparing data on age (Table S1), which is obviously characterized by the absence of equality of variances, should be explained.

T-test was also used when comparing data, presented in Table S2, with a number of observations in the group = 4. Please confirm that, with such few observations, the data had a normal distribution without variance differences.

*Reviewer #2 (Recommendations for the authors):*

In this article, the role of B cells in immune cell activation in PCOS was determined. This is a very exciting paper with novel information in an area of growing interest and highly clinically relevant. One of the clear strengths of this manuscript is that it combines basic research with clinical studies in PCOS women. The main findings of the manuscript are:

– Androgens are associated with altered B cell frequencies and immunoglobulin M increase in women with PCOS.

– Transfer of human-derived IgG antibodies results in increased body weight in WT female mice but not glucose impairment.

– Altered B cell frequencies are replicated in a DHT-induced PCOS-like mouse model and seen in reproductive, metabolic, and immunological tissues.

– DHT-induced PCOS-like mice show a distinct IgG profile.

– B cell transfer from DHT-induced PCOS-like mice into B cell-deficient mice does not induce a PCOS-like phenotype.

– B cell deficiency does not protect from the induction of a PCOS-like phenotype by DHT exposure

Experiments were carried with adequate control.

I have the following suggestions that the authors may consider.

Please clarify the number of PCOS participants for IgG donor experiments, in figure looks like n of 5 while in Table S2 only 4. Regardless, the low number of participants should be included in the weakness section of the discussion. The lack of effects on the full manifestations could be due to a low number of participants and also in those women the level of androgens was normal.

IgA was decreased in PCOS, but was not significant, perhaps due to the low number of subjects.

Figure 7, was a group treated with flutamide in this cohort?

Clinical characteristics of the PCOS participants should be moved to the main result section.

The statistical section mentions 40 participants. This is confusing as two sets of PCOS were used.

---

## [Author Response]

Essential revisions:Please, consider this list of essential issues that could help you to improve the clarity and presentation of your work:1) Check the number of subjects at each stage of study and report it correctly in the tables and figures.

We thank you for highlighting that our reporting of the number of subjects was unclear. We have now referenced the number of subjects within all legends of individual figures and tables and hope to add clarity also for future readers' sample sizes.

2) Explain the reasons for the use of T-test criteria for some comparisons presented in Tables S1-2.

In re-analyzing the test assumptions for the data in Tables S1-2 (now included in the main file as Table 1 and 2), we acknowledge that the T-Test may not be the most fitting for the comparisons presented. Hence, we used the non-parametric Mann-Whitney U-test for statistical analyses in Tables 1-2.

3) Discuss the limitations of the study considering sources of potential bias.

We agree with the editors that this aspect needs to be clearly addressed for the readers within the discussion. One limitation was the number of participants recruited throughout the pandemic years, particularly for the extraction of purified IgG. We have now highlighted within the discussion our main concerns and stated as follows:

“A limitation of the study that should be noted is that the number of PCOS donors for IgG transfer was low with no significant difference in circulating androgen values, potentially affecting the biological effects resulting from the transfer. Based on the availability of purified serum, the number of recipient mice was also limited potentially leading to a loss of precision and statistical power in the case of missing values in the outcome.”

Reviewer #1 (Recommendations for the authors):The study is well-designed, the methods are convincing. Generally, results are based on the compelling strength of evidence, but there are some remarks regarding the data presentation.The total number of screened women should be presented in the Material and Methods section (page 13), because n=15 is the final number of recruited women with PCOS.

Indeed, this is of importance and the number of included participants has been clarified in Material and Methods (Page 13-14) and is also given in tables and legends of figures.

It looks reasonable to present the number of observations in all the Figures.

We agree with the reviewer and to assist readers we have added the number of subjects for each stage of the study in the figure legends.

Please, comment on the significant age difference between women from the PCOS and control groups, which may cause some biases.At the same time, the use of the T-test when comparing data on age (Table S1), which is obviously characterized by the absence of equality of variances, should be explained.

As requested by the reviewer, we expanded the Discussion section with an indepth consideration regarding the potential effects stemming from the age differences between cases and control in the first cohort. We believe this facilitates a better understanding of double negative (DN) B cells in PCOS for the readers.

We clarified their nature as notably “age-associated” (ABCs) immunosenescent B cells and stated the following (page 9):

“This [finding] occurred independently of the significant age difference noted between the two cohorts both randomly recruited, as PCOS affected women were younger than control participants. These are in fact age-associated tissuebased atypical memory B cells which represent a low prevalence population, generally noted in the elderly, positively impacting immunosenescence. They have been however described in previous studies as autoantibody enriched, poised for plasma cell differentiation, and may over-accumulate prematurely in chronic infections, autoimmune diseases, and immunodeficiencies thus playing a role in the regulation of humoral responses”. In the following paragraph we further highlighted this aspect in our interpretation based on the results that a condition-specific antigenic stimulus deriving from PCOS inflammatory state “which is most likely independent of age, causing rearrangement of the humoral immune system” (page 10).

T-test was also used when comparing data, presented in Table S2, with a number of observations in the group = 4. Please confirm that, with such few observations, the data had a normal distribution without variance differences.

In re-analyzing the test assumptions for the data in Tables S1-2 (now included in the main file as Table 1-2), we acknowledge that the T-Test may not be the most fitting for the comparisons presented. Instead, we now use the non-parametric Mann-Whitney U-test for statistical analyses in Tables 1-2.

Reviewer #2 (Recommendations for the authors):I have the following suggestions that the authors may consider.Please clarify the number of PCOS participants for IgG donor experiments, in figure looks like n of 5 while in Table S2 only 4. Regardless, the low number of participants should be included in the weakness section of the discussion. The lack of effects on the full manifestations could be due to a low number of participants and also in those women the level of androgens was normal.

We apologize for the lack of clarity in presenting this aspect of the data. Figure 5 describes the experiment where we investigate androgen-mediated regulation of B cell phenotypes, DN B memory cells as well as circulating antibody titers, in tissues other than blood, in the well-established peripubertal DHT-induced PCOS mouse model.

The donor experiment is presented in Figure 2 and we have now added the exact number of IgG donors as well as the number of recipient mice described.

Following the reviewer`s recommendation, we have proceeded by noting the clinical characteristics of the IgG donors firstly within the results (page 6) which is stated as following:

“Both groups of donors were age-homogeneous with no significant differences in BMI or circulating androgen levels” as well as within the discussion (page 10) describing a relevant limitation of this study´s outcome by stating as follows:

“A limitation of the study that should here be noted is that the number of PCOS donors for IgG transfer was ultimately quite low with normal androgen values, potentially affecting the biological effects resulting from the transfer. Based on the availability of purified serum, the number of recipient mice was also restricted.”

IgA was decreased in PCOS, but was not significant, perhaps due to the low number of subjects.

We agree with the reviewer that the low number of participants decreased the statistical power, although an interpretation of the overall decrease in IgA may not be as equally straightforward based on our findings. It remains unclear if the decrease in IgA is indeed a distinct humoral profile in PCOS possibly due to defective germinal center reactions and the extrafollicular expansion of unique IgM^+^ DN B cells, or rather a compensatory mechanism to the increase of IgM.

Figure 7, was a group treated with flutamide in this cohort?

No, and indeed it may have been a more elegant manner to complete the results. This aspect of the study was designed to challenge the main hypothesis that a complete lack of B cells would be protective. Based on the previous wild-type experiments we reasoned to have demonstrated sufficiently the effects of androgen receptor activation in an attempt to follow the 3R principles, optimizing the amount of information with fewer animals.

Clinical characteristics of the PCOS participants should be moved to the main result section.

The clinical characteristics has now been included in the main manuscript as Table 1 and 2.

The statistical section mentions 40 participants. This is confusing as two sets of PCOS were used.

Indeed, this is of importance and the number of included participants has been clarified in Material and Methods (Page 13-14) and is given in tables and legends of figures. Regarding samples size estimation we have clarified as follows (Statistics; page 19):

“For the first cohort, the human monocentric case-control explanatory study based on B cell frequencies […] we aimed for a total sample of 40 subjects, 20 PCOS and 20 controls”; “For the second study cohort, for the transfer of purified IgG, a predefined sample size was not assigned.”